# Membrane potential dynamics of excitatory and inhibitory neurons in mouse barrel cortex during active whisker sensing

Taro Kiritani[¤a], Aurélie Pala[¤b], Célia Gasselin, Sylvain Crochet[‡]*, Carl C. H. Petersen[‡]*

Laboratory of Sensory Processing, Brain Mind Institute, Faculty of Life Sciences, École Polytechnique Fédérale de Lausanne (EPFL), Lausanne, Switzerland

¤a Current address: ExaWizards Inc., Tokyo, Japan
¤b Current address: Department of Biology, Emory University, Atlanta, GA, United States of America
‡ SC and CCHP are Joint Senior Authors on this work.
* sylvain.crochet@epfl.ch (SC); carl.petersen@epfl.ch (CCHP)

**Data Availability Statement:** The complete data set and Matlab analysis code are freely available at the CERN database Zenodo with doi: https://doi.org/10.5281/zenodo.7833080.

## Abstract

Neocortical neurons can increasingly be divided into well-defined classes, but their activity patterns during quantified behavior remain to be fully determined. Here, we obtained membrane potential recordings from various classes of excitatory and inhibitory neurons located across different cortical depths in the primary whisker somatosensory barrel cortex of awake head-restrained mice during quiet wakefulness, free whisking and active touch. Excitatory neurons, especially those located superficially, were hyperpolarized with low action potential firing rates relative to inhibitory neurons. Parvalbumin-expressing inhibitory neurons on average fired at the highest rates, responding strongly and rapidly to whisker touch. Vasoactive intestinal peptide-expressing inhibitory neurons were excited during whisking, but responded to active touch only after a delay. Somatostatin-expressing inhibitory neurons had the smallest membrane potential fluctuations and exhibited hyperpolarising responses at whisking onset for superficial, but not deep, neurons. Interestingly, rapid repetitive whisker touch evoked excitatory responses in somatostatin-expressing inhibitory neurons, but not when the intercontact interval was long. Our analyses suggest that distinct genetically-defined classes of neurons at different subpial depths have differential activity patterns depending upon behavioral state providing a basis for constraining future computational models of neocortical function.

## Introduction

The neocortex contributes importantly to sensory perception, cognition and motor control. Sensory cortical areas are driven by specific thalamic input arranged in maps of cortical columns organised according to retinotopy for vision, tonotopy for hearing and somatotopy for touch. Each cortical column spans the thickness of the neocortex and can be divided into layers, with the specific thalamic axonal innervation density being highest in layer 4 (L4). Many different classes of neurons are found throughout the different layers and regions of the

**Funding:** This work was supported by project grant 310030_146252 from Swiss National Science Foundation (https://www.snf.ch/en) (CCHP) and Advanced grant 293660 from the European Research Council (https://erc.europa.eu) (CCHP). The funders had no role in study design, data collection and analysis, decision to publish, or preparation of the manuscript. There was no additional external funding received for this study.

**Competing interests:** The authors have declared that no competing interests exist.

neocortex, and, in the mouse, these neuronal classes are increasingly well-defined through gene expression patterns, morphology, connectivity and electrophysiological properties [1–16]. As the cellular composition of the neocortex begins to be characterized in detail, it becomes equally important to investigate the activity patterns of the various cell classes in different cortical depths. In functional studies, the excitatory neurons releasing glutamate are typically subdivided according to the laminar location of the cell body or long-range projection targets, whereas the inhibitory neurons releasing GABA are typically subdivided into four largely-non-overlapping molecularly-defined classes expressing parvalbumin (PV), vasoactive intestinal peptide (VIP), somatostatin (SST) or 5HT3A-non-VIP [8, 13]. Calcium imaging and extracellular electrophysiological recordings have been widely used to investigate action potential-related signals in different classes of cortical neurons, but for deeper mechanistic understanding it is necessary to uncover the underlying subthreshold membrane potential ($V_m$) fluctuations which drive action potential (AP) firing. Whole-cell $V_m$ recordings provide the highest signal-to-noise ratio measurements of neuronal activity, and here we target whole-cell recordings to fluorescent genetically-defined classes of PV, VIP and SST GABAergic neurons, as well as excitatory neurons, in head-restrained behaving mice focusing on defining their $V_m$ dynamics during active whisker sensation [17].

The whisker representation in mouse primary somatosensory cortex has a remarkably clear somatotopic organization, with each mystacial whisker being individually mapped onto anatomically-defined cortical barrel columns [18, 19]. Sensorimotor processing in the barrel cortex contributes to whisker-dependent touch detection, object localization, texture discrimination and shape discrimination [20, 21]. Mice actively obtain tactile sensory information from their whiskers by moving them back-and-forth to scan their immediate facial environment. When the whisker contacts an object, forces are exerted on the whisker follicle and sensory nerve endings of trigeminal neurons are excited by touch. Sensory signals from the whiskers are transmitted in the form of APs releasing glutamate upon second-order neurons in the trigeminal brain stem, which in turn project to many downstream brain areas [22]. The lemniscal pathway involving the principal trigeminal nucleus and the ventral posterior medial (VPM) thalamic nucleus is the primary route for whisker sensory information to arrive in the barrel cortex [23]. The VPM differentially innervates neurons according to layer and cell class, with excitatory and PV neurons in L4 receiving the biggest direct glutamatergic input [24]. Excitatory neurons in L4 provide columnar synaptic input onto neurons in all other cortical layers [25], thus forming the beginning of cortical sensory processing according to the canonical microcircuit model [26]. However, there is also substantial VPM input to L3 [24], L5 [27] and L6 [28]. Although a small number of $V_m$ recordings from genetically-defined deeper-lying neurons in barrel cortex of actively sensing mice have previously been reported using optogenetic tagging [29, 30], most previous $V_m$ studies of different classes of GABAergic neurons in mouse barrel cortex during behavior have largely focused on superficial neurons located in the upper ~300 μm of the neocortex since these are more-easily targeted through two-photon microscopy. Such measurements in awake mice have revealed interesting differences in $V_m$ dynamics comparing PV, VIP and SST neurons [31–33], as well as differences across excitatory neurons in terms of cortical depth [34] and long-range projections [35]. In this study, we extend current knowledge by investigating $V_m$ dynamics across a greater range of depths including two-photon targeted whole-cell recordings across the upper ~600 μm of the neocortex from excitatory, PV, VIP and SST neurons during active whisker sensation. This is important because previous extracellular measurements of cell class-specific AP firing in barrel cortex have revealed prominent differences depending upon the depth of the cell body relative to the pial surface [29, 30, 36]. Consistent with the extracellular measurements, our intracellular data reveal differences in $V_m$ comparing across these 4 different cell classes and across

cortical depth during quiet states without whisker movement, free whisking in air without object touch, and during active whisker touch of an object.

## Materials and methods

### Animal preparation, surgery and habituation to head fixation

All experiments were conducted in accordance with the Swiss Federal Veterinary Office under authorization VD-1628 from the Office of Veterinary Affairs of the Canton of Vaud, Switzerland. Adult (4–8 week-old) male and female mice were implanted under deep isoflurane anesthesia with a light metal-post as previously described [37]. After full recovery from surgery, mice were gradually habituated to head-fixation and accustomed to the 2-photon microscope set-up across a few sessions of increasing duration. All whiskers, except the right C2 whisker, were trimmed.

The following transgenic mouse lines were used in this study: C57BL6J (Janvier, France); B6;C3-Tg(Scnn1a-cre)3Aibs/J (Scnn1a-Cre; JAX: 009613) [38]; B6(Cg)-Etv1$^{tm1.1(cre/ERT2)Zjh}$/J (Etv1-Cre; JAX: 013048) [39]; Vip$^{tm1(cre)Zjh}$/J (VIP-Cre; JAX: 010908) [39]; B6;129P2-*Pvalb*$^{tm1(cre)Arbr}$/J (PV-Cre; JAX: 008069) [40]; Sst$^{tm2.1(cre)Zjh}$/J (SST-Cre; JAX: 013044) [39]; B6.Cg-Gt (ROSA)26Sor$^{tm9(CAG-tdTomato)Hze}$/J (LSL-tdTomato; JAX: 007909) [38].

### Electrophysiology

The location of the C2 barrel column in the left hemisphere was functionally determined using intrinsic optical signal imaging, as previously described [25, 41]. A few hours before the recording session, a craniotomy (1–2 mm) was opened over the right-C2 whisker barrel column and a 3 mm coverslip was halved and glued over roughly two-thirds of the craniotomy allowing for optical access for 2-photon imaging and for the insertion of the glass pipette for patch-clamp whole-cell recording. After 2–3 hours of recovery, the mouse was head-fixed under the 2-photon microscope and targeted patching was performed as previously described [42].

Whole-cell recordings were targeted to fluorescent neurons visualized with a two-photon microscope. The scanning system of the custom-made two-photon microscope consisted of a galvo-resonance mirror pair and fluorescence was excited with a femtosecond pulsed tuneable infrared laser focused into the cortex with a water immersion objective. The acquisition and imaging hardware was controlled by a Matlab-based software (ScanImage SI5) [43]. Green and red fluorescence were detected using GaAsP photosensor modules after passing through an infrared blocker. Photocurrents were amplified and digitized with an A/D board with a frame acquisition rate of 30 Hz at a resolution of 512x512 pixels.

Whole-cell recording pipettes were filled with an intracellular solution containing (in mM): 135 K-gluconate, 4 KCl, 4 Mg-ATP, 10 Na$_2$-phosphocreatine, 0.3 Na$_3$-GTP, and 10 HEPES (pH 7.3, 280 mOsmol/l). To target tdTomato-labelled neurons 1–20 μM of Alexa488 or 1 μM SeTau-647 was included in the pipette solution. Neurons were recorded in the current-clamp configuration using a Multiclamp 700B amplifier (Molecular Devices). Borosilicate patch pipettes with resistance of 5–7 MΩ were used. Electrophysiological data were low-pass Bessel filtered at 10 kHz and digitized at 40 kHz. Recordings were obtained without injecting current and membrane potential measurements were not corrected for the liquid junction potential. The depth of the recorded cell was estimated as the difference in the focal distance between the pia and the cell body visualized with the 2-photon microscope.

## Monitoring whisker movements

To monitor whisker angular position and contacts between the whisker and the object, we filmed the right C2 whisker during 20–60 second sweeps of membrane potential recording. All whiskers except for C2 were trimmed before the recording session. The behavioral images were synchronized to the electrophysiological recording through TTL pulses. In some recording sweeps, a small object with a vertical edge was introduced in the path of the right C2 whisker in such a way that the mouse could actively make contacts with the object by protracting its whisker (active touch). Whisker movements and whisker-object contacts were quantified off-line through analysis of the high-speed video and additionally through signals from a piezo sensor used for the object [34, 44].

## Anatomy

After transcardial perfusion and postfixation for 2–4 hr using paraformaldehyde (4%, in 0.1 M phosphate buffer [PB], pH 7.3–7.4), a tangential section (800 μm to 2 mm thick) containing the C2 barrel was sliced. The brain section was washed and kept in PBS before histological staining and tissue clearing. The tissue was cleared in CUBIC-mount [45] for more than 48 hours, and then sealed in a chamber filled with CUBIC-mount with #1.5 cover glass. We used a confocal microscope to image the C2 barrel column. tdTomato fluorescence signal was obtained with excitation at 555 nm or 561 nm, and emission 575–640 nm. In Scnn1a-tdTomato, and PV-tdTomato mice, the morphology of L4 barrels was visualized through the bright tdTomato fluorescence from the barrels. In SST and VIP mice, the tdTomato signal was not strongly correlated with the shape of barrels, and in these mice, we instead used intrinsic green autofluorescence (excitation: 488 nm, emission: 515–565 nm) to visualize barrels.

## Data analysis

**Quantification of whisker movement.** Behavioural states of mice were classified as quiet or whisking based on the angular speed of the right C2 whisker. To select periods of quiet wakefulness, the angular whisker position signal was first low-pass filtered at 40 Hz. Then the power of the first derivative of the filtered whisker angle was used to detect high-velocity whisking events above 25,000 $deg^2 \cdot s^{-2}$. Consecutive whisking events separated by less than 500 ms were first fused, then remaining short and isolated whisking events of less than 10 ms were discarded. Only periods with no whisking activity left were considered as quiet wakefulness. To select periods of whisking, we applied the same procedure but used a higher threshold to detect whisking events (75,000 $deg^2 \cdot s^{-2}$), a shorter time window for fusing consecutive whisking events (400 ms) and we removed isolated whisking events shorter than 100 ms. As a result, periods with intermediate whisker movement activity were classified neither as quiet nor as whisking.

For the analysis of the transition at whisking onset, we selected whisking periods lasting at least 300 ms, preceded by at least 1 s without any whisking periods and devoid of any contact between the whisker and the object. Only recordings for which we obtained at least 5 transitions were included in the analysis.

For the analysis of steady-state quiet and whisking periods, we cut the $V_m$ recordings in 2 s time-windows of homogeneous quiet or whisking periods. Only recordings that contained at least 3 time-windows for both quiet and whisking states were included in the analysis.

To analyse the fast correlation between $V_m$ and whisker angular position during whisking, we first extracted single whisking cycles by filtering the whisker angle between 5 and 25 Hz, then selected whisking cycles of amplitude higher than 15 deg with no contact with the object 500 ms before or after. We extracted the phase of each selected whisking cycle using the Hilbert

transform. Then we divided each whisking cycle into 100 phase bins and reconstructed the mean phase-aligned $V_m$ trajectory similarly to Hill et al. (2011) and Sreenivasan et al. (2016) [46, 47]. Only recordings with at least 10 whisking cycles were included in the analysis.

For active contact with the object, we first selected contacts with no contact 100 ms before and only included recordings with at least 5 contacts. To assess the impact of inter-contact interval (ICI), we split the contacts into long ICI (no contact within preceding 100 ms) and short ICI (preceding contact within 80 ms) and included only recordings with at least 3 contacts in each category.

**Subthreshold and suprathreshold analysis of $V_m$.** The $V_m$ signal was decomposed into suprathreshold (AP) and subthreshold activity. APs were first detected as events with high $V_m$ first derivative ($>5$–$40$ mV.ms$^{-1}$) and high amplitude ($> 5$ mV). AP times were defined at the peak of the AP. AP threshold was defined as the $V_m$ at the time $V_m$ crossed the threshold of $5$–$40$ mV.ms$^{-1}$ just before the AP peak. The subthreshold $V_m$ activity was obtained by cutting the APs from the $V_m$ recording by interpolating a linear segment between the AP threshold and the time point after the peak of the AP that first met the following conditions: 1) the $V_m$ returns to AP threshold level or 2) the $V_m$ 1$^{st}$ derivative becomes positive, within the next 15 ms following the AP peak.

To compute AP parameters, we selected APs during periods of quiet wakefulness, without any AP 50 ms before and after, and with amplitude higher than 20 mV. We computed the mean AP waveform for each neuron and defined the AP duration as the mean duration at half amplitude computed from the averaged AP waveform.

To compute subthreshold $V_m$ properties, we selected 2 s time-windows of quiet wakefulness and computed the mean, the SD and the FFT of the $V_m$ for each 2 s time windows and then averaged across the different time windows for each neuron.

**Comparison across cellular depths and cortical layers.** To assess the impact of the position of the recorded neurons in the cortical column, we correlated the physiological measurements with the recording depth or the estimated cortical layer. Because we did not recover the anatomy of all the recorded neurons, we used the recording depth to assign each neuron to a layer using the following boundaries: L2-L3, 250 μm; L3-L4, 400 μm; L4-L5, 600 μm. These boundaries are consistent with previously published anatomical measurements [25] and we found a good match between the assignment of cortical layers based on those criteria and the actual cortical layers for the recorded neurons that were recovered and anatomically identified.

**Statistics.** Sample sizes were not calculated at the design of the study, and they reflect the probability of obtaining $V_m$ recordings from the different cell classes. No randomization or blinding was implemented.

Averaged traces and FFTs are shown as mean ± SEM. Grouped data are plotted as mean ± SD. Statistical significance was assessed using a Wilcoxon signed-rank test against the null hypothesis. Comparisons between the four cell classes, or between cortical layers, were assessed by performing a Kruskal-Wallis test followed, when appropriate, by a Tukey-Kramer multiple comparison test. Linear correlation between two parameters was tested with a Pearson correlation test. The dependence of a measured parameter with cell depth was tested using the non-parametric Spearman correlation test.

## Results

### Whole-cell $V_m$ recordings of excitatory and inhibitory neurons in mouse barrel cortex

Under visual control offered by a two-photon microscope, we targeted whole-cell recordings to fluorescently-labelled neurons in the C2 barrel column of the left hemisphere of awake

head-restrained mice while filming whisker movements and, at times, presenting an object for the mouse to palpate with its right C2 whisker. In some attempts, we obtained whole-cell recordings from nearby unlabeled neurons, which in our analyses were assigned to be excitatory neurons. We recorded from excitatory (Fig 1A), PV (Fig 1B), VIP (Fig 1C) and SST (Fig 1D) neurons. In addition to newly obtained data (n = 192 neurons), here we include new analysis of previously published data from 10 PV neurons and 10 SST neurons [33]; as well as new analysis of 28 previously published anatomically-identified excitatory neurons [34]. Altogether the dataset analysed in this study contains 93 excitatory neurons, 49 PV neurons, 25 VIP neurons and 73 SST neurons distributed largely from 100 to 600 μm below the pial surface (Fig 1E). The neurons were recorded in various strains of mice: C57BL6J (n = 28 mice); Scnn1a-Cre x LSL-tdTomato (n = 9 mice); Etv1-Cre x LSL-tdTomato (n = 10 mice); PV-Cre x LSL-tdTomato (n = 35 mice); VIP-Cre x LSL-tdTomato (n = 16 mice); and SST-Cre x LSL-tdTomato (n = 56 mice). Because recordings under visual control become more difficult to carry out deeper in the brain, our data sample is skewed towards superficial neurons compared to the overall cell class-specific distributions of the cell bodies of GABAergic neurons in the C2 barrel column (Fig 1F and S1–S4 Videos).

## $V_m$ dynamics during quiet wakefulness

We first analysed $V_m$ dynamics during quiet wakefulness, which was defined to be periods in which the mouse did not move its whisker (see **Materials and methods**). Typically, movements of other body parts are also accompanied by whisker movement, so in the quiet state it is likely that there is very little movement of the mouse altogether, defining what one might consider as a 'ground state' of neuronal activity in the awake somatosensory cortex. We investigated both suprathreshold and subthreshold $V_m$ activity across the four classes of neurons (Fig 2A). AP waveforms differed in their full-width at half-maximum amplitude across all classes of neurons, except excitatory and VIP neurons which were not significantly different (Fig 2B). On average PV neurons had the shortest APs and excitatory neurons had the longest APs. The AP durations (mean ± SD) were: excitatory 1.08 ± 0.42 ms (n = 88 cells); PV 0.35 ± 0.10 ms (n = 46 cells); VIP 0.80 ± 0.32 ms (n = 25 cells); and SST 0.57 ± 0.20 ms (n = 67 cells). Interestingly, the AP duration for excitatory neurons was minimal around 400 μm below the pial surface (S1A and S2A Figs). AP firing rates also differed across all classes of neurons, except VIP and SST neurons which were not significantly different (Fig 2C). On average, PV cells fired at the highest rates and excitatory cells at the lowest rates. The AP firing rates were: excitatory 1.7 ± 2.2 Hz (n = 93 cells); PV 27.3 ± 20.0 Hz (n = 49 cells); VIP 10.2 ± 10.2 Hz (n = 25 cells); and SST 6.5 ± 6.2 Hz (n = 73 cells). The firing rate of excitatory neurons showed a positive correlation with recording depth whereas a negative correlation was found for PV neurons (S1B and S2B Figs). Excitatory neurons on average had a more hyperpolarised mean subthreshold $V_m$ compared to inhibitory neurons, but the mean $V_m$ of PV, VIP and SST neurons were not statistically different (Fig 2D). The mean $V_m$ of the four different cell classes were: excitatory -58.7 ± 5.9 mV (n = 87 cells); PV -52.7 ± 3.9 mV (n = 49 cells); VIP -51.8 ± 4.2 mV (n = 25 cells); and SST -51.6 ± 4.7 mV (n = 73 cells). The mean $V_m$ of excitatory neurons was positively correlated with cell depth (S3A and S4A Figs). The standard deviation of subthreshold $V_m$ across time differed across all classes of neurons, except VIP and SST neurons which were not significantly different (Fig 2E). PV cells had the highest $V_m$ standard deviation while SST neurons had the lowest. The standard deviation of $V_m$ of the four different cell classes were: excitatory 5.6 ± 1.3 mV (n = 87 cells); PV 6.4 ± 1.6 mV (n = 49 cells); VIP 5.2 ± 2.0 mV (n = 25 cells); and SST 4.0 ± 1.0 mV (n = 73 cells). We also computed the fast Fourier transform (FFT) to examine the frequency spectra of the $V_m$ fluctuations, finding that SST cells

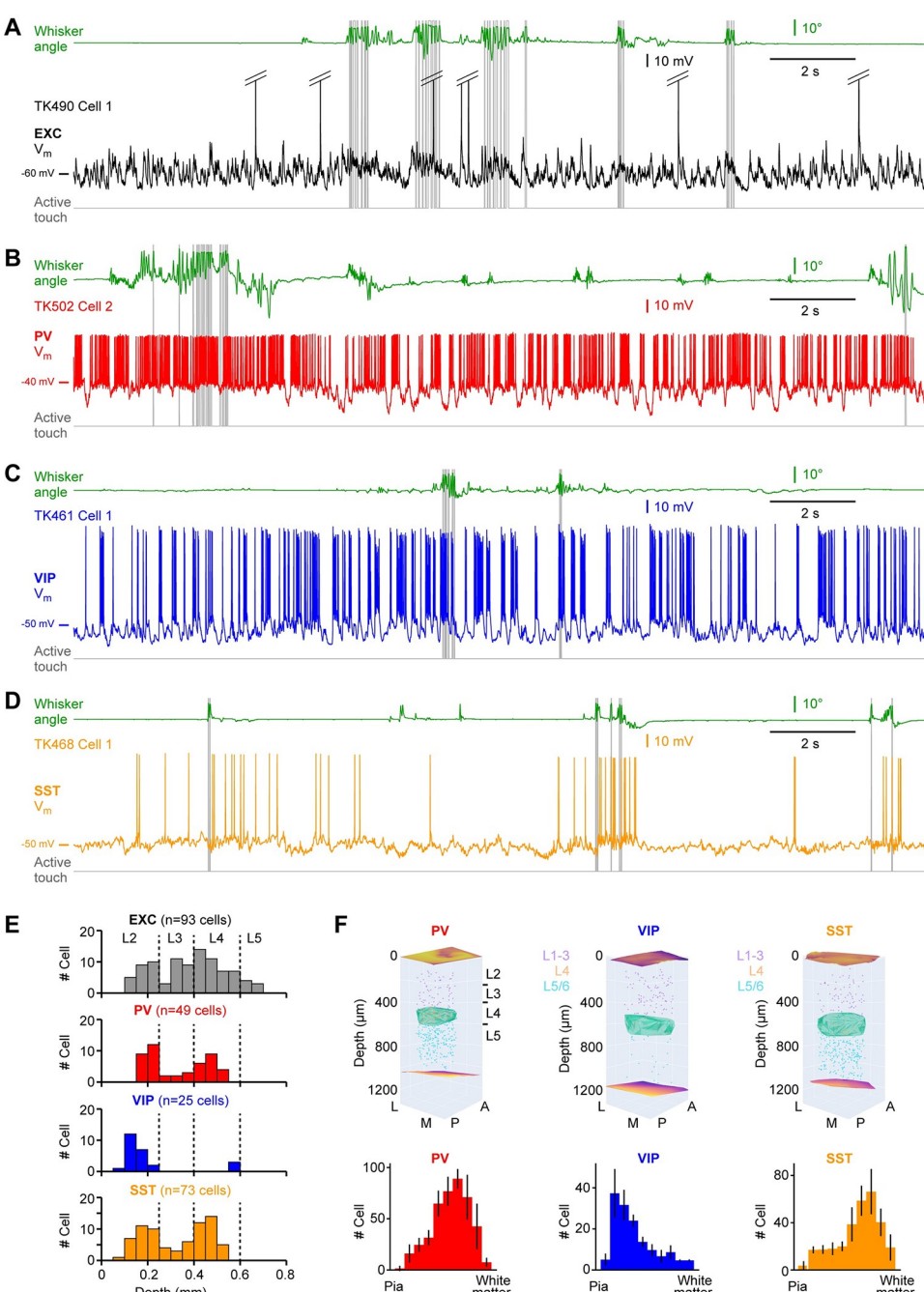

**Fig 1. Example membrane potential recordings and cortical depth distributions of cell classes.** (**A**) Membrane potential (V$_m$) recording from an excitatory neuron recorded at 375 μm below the pial surface. From top to bottom: right C2 whisker angular position (green); V$_m$ (black, APs are truncated); active contacts between the C2 whisker and the object (grey). (**B**) Same as A, but for a PV-expressing GABAergic neuron recorded at 185 μm below the pial surface (V$_m$, red). (**C**) Same as A, but for a VIP-expressing GABAergic neuron recorded at 227 μm below the pial surface (V$_m$, blue). (**D**) Same as A, but for an SST-expressing GABAergic neuron recorded at 207 μm below the pial surface (V$_m$, orange). (**E**) Distribution of the estimated cell body depths for each recorded neuron according to cell class. Dashed lines indicate depth boundaries used to define cortical layers. (**F**) Anatomical reconstruction of the cell body locations of GABAergic neurons within the C2 barrel column (top) in three example brains from which we did not record, with the C2 barrel in layer 4 colored green. The distributions of PV-expressing (n = 4 mice), VIP-expressing (n = 3 mice) and SST-expressing (n = 3 mice) neurons are quantified across mice along the depth of the cortical column (below) with the histogram indicating mean ± SD.

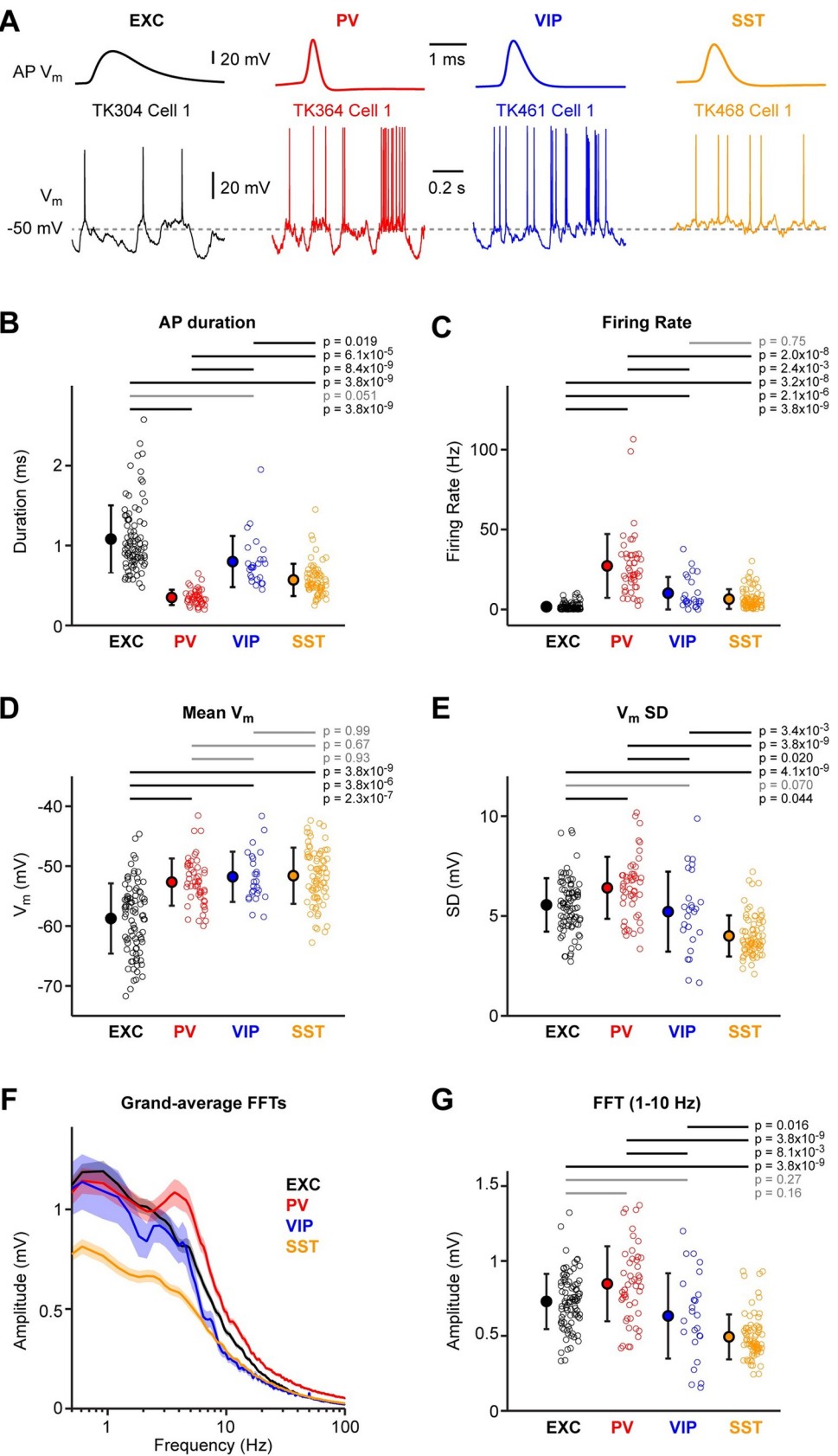

**Fig 2. Supra- and sub-threshold membrane potential fluctuations during quiet wakefulness.** (**A**) Example $V_m$ recordings from excitatory (EXC), PV-expressing, VIP-expressing and SST-expressing neurons, during 1 s of quiet wakefulness. Averaged action potentials (APs) are shown above on an expanded timescale. (**B**) Mean AP duration for the four cell classes. Open circles show individual neuron values. Filled circles with error bars show mean ± SD. Statistical differences between cell classes were computed using a Kruskal-Wallis test ($p = 3.2 \times 10^{-32}$) followed by a Tukey-Kramer multiple comparison test. (**C**) Same as B, but for the mean firing rate during (Kruskal-Wallis test, $p = 3.3 \times 10^{-29}$). (**D**) Same as B, but for the mean subthreshold $V_m$ (Kruskal-Wallis test, $p = 8.1 \times 10^{-15}$). (**E**) Same as B, but for the mean standard deviation (SD) of the subthreshold $V_m$ (Kruskal-Wallis test, $p = 1.9 \times 10^{-16}$). (**F**) Grand-average FFTs computed from the subthreshold $V_m$ for each cell class. (**G**) Same as B, but for the mean 1–10 Hz FFT amplitude of the subthreshold $V_m$ (Kruskal-Wallis test, $p = 1.7 \times 10^{-16}$).

appeared to have the lowest amplitude of slow $V_m$ dynamics (Fig 2F). Quantified across the 1–10 Hz frequency range, all groups of neurons showed different amplitude $V_m$ fluctuations, except excitatory vs PV neurons and excitatory vs VIP neurons which were not significantly different (Fig 2G). On average, PV cells had the largest 1–10 Hz $V_m$ fluctuations while SST neurons had the smallest. The $V_m$ standard deviation and amplitude of 1–10 Hz fluctuations were larger in superficial PV and SST cells, compared to deeper cells (S3B, S3C and S4B, S4C Figs).

Examining the correlation between different parameters of the $V_m$ dynamics, we found a negative correlation between the firing rate and AP duration. Plotting the firing rate against the AP duration for all neurons, showed that PV and excitatory cells form two non-overlapping clusters, whereas VIP and SST cells are distributed in between and overlap with both PV and excitatory neurons (S5A Fig). We also found that the firing rate was significantly correlated with the mean $V_m$, and to a lesser extent with the standard deviation of $V_m$, but not with the AP threshold (S5B–S5D Fig). The initiation of APs was on average preceded by a large $V_m$ depolarization in the preceding 20 ms, with the strongest pre-spike depolarizations found in excitatory neurons (S6 Fig), as previously observed [42].

The four different classes of neurons examined in this study therefore have different suprathreshold and subthreshold $V_m$ dynamics during quiet wakefulness.

## $V_m$ dynamics at the onset of whisking

We next examined the $V_m$ changes that occur at the onset of whisking as the mouse transitions from quiet wakefulness to active whisker sensing without object contacts (free whisking) (Fig 3A). All four classes of neurons showed changes in $V_m$ dynamics at the onset of whisking (Fig 3B). Excitatory, PV and VIP neurons depolarized, whereas SST neurons transiently hyperpolarized at the onset of whisking (Fig 3B). The depolarization of excitatory, PV and VIP neurons was not significantly different between these cell classes, but the hyperpolarization of SST neurons differed significantly from the depolarization of excitatory, PV and VIP neurons (Fig 3C). The changes in $V_m$ at the onset of whisking for the four different cell classes were: excitatory 1.0 ± 3.0 mV (n = 73 cells); PV 2.1 ± 2.4 mV (n = 40 cells); VIP 2.6 ± 3.7 mV (n = 20 cells); and SST -1.4 ± 2.5 mV (n = 55 cells). AP firing increased for PV and VIP neurons at the onset of whisking, but not for excitatory or SST neurons (Fig 3D). The increased firing rate of PV and VIP neurons did not differ significantly, but the increased firing rate of both PV and VIP neurons was significantly larger than excitatory and SST neurons (Fig 3D). The changes in AP firing rates at the onset of whisking were: excitatory 0.2 ± 2.9 Hz (n = 73 cells); PV 10.6 ± 16.7 Hz (n = 40 cells); VIP 6.8 ± 9.4 Hz (n = 20 cells); and SST 0.0 ± 6.4 Hz (n = 55 cells). For PV cells we found a weak correlation of the whisking onset-related increase in AP firing with cell depth (S7B Fig). For all cell classes we observed significant linear correlation between the change in firing rate and the change in $V_m$ at whisking onset (S7C Fig).

Interestingly, the hyperpolarizing $V_m$ observed for SST neurons at whisking onset was prominent for superficial neurons, but absent on average for deeper SST neurons (Fig 3E). We

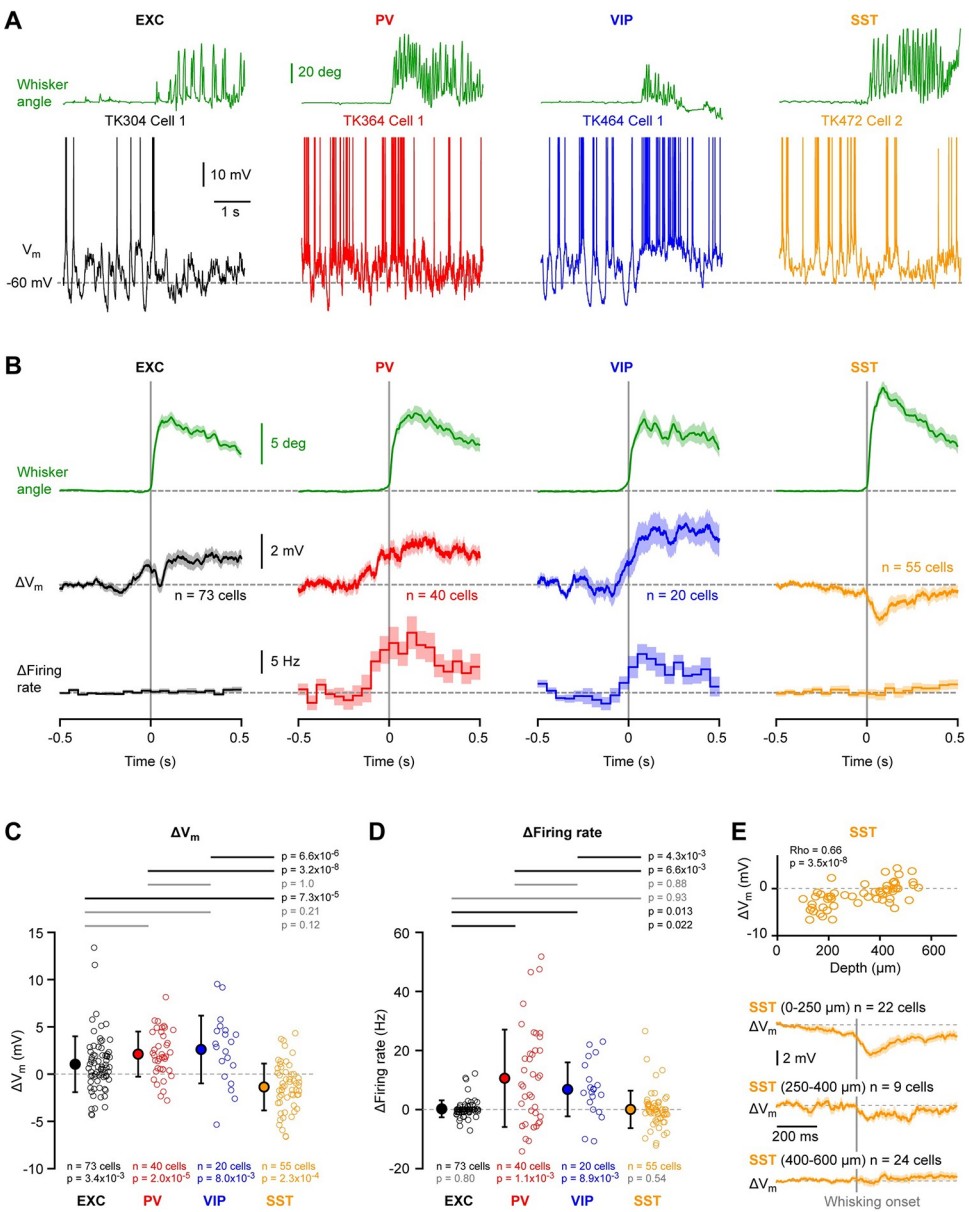

**Fig 3. Supra- and sub-threshold membrane potential changes at the onset of whisking without object contact. (A)** Example $V_m$ recordings from excitatory (EXC), PV-expressing, VIP-expressing and SST-expressing neurons during the transition from quiet wakefulness to active whisking without object contacts. The angular whisker position (green) is shown above and the $V_m$ below (APs truncated). **(B)** From top to bottom: grand-average whisker angle, $V_m$ and firing rate aligned to whisking onset at time = 0 s computed for each cell class. **(C)** Mean change in $V_m$ at whisking onset (0–200 ms) for the four cell classes. Open circles show individual neuron values. Filled circles with error bars show mean ± SD. Statistical differences for the change in membrane potential comparing quiet and whisking were computed using a Wilcoxon signed rank test for each cell class (shown below the data points). Statistical differences between cell classes were computed using Kruskal-Wallis test (p = 1.1x10$^{-9}$) followed by a Tukey-Kramer multiple comparison test (shown above the graph). **(D)** Same as C, but for the change in firing rate (Kruskal-Wallis test, p = 1.9x10$^{-4}$). **(E)** $V_m$ change at whisking onset in SST-expressing neurons vs cell depth showed a significant positive correlation (Spearman test: Rho = 0.66 with p = 3.5x10$^{-8}$) (*above*). Open circles represent individual neurons. Grand-average $V_m$ at whisking onset for SST-expressing neurons recorded between 0–250 μm (L2), 250–400 μm (L3) and 400–600 μm (L4) below the pial surface (*below*).

found a significant positive correlation between the change in $V_m$ and in firing rate at the onset of whisking with the depth of the SST neurons (Rho = 0.66, p = 3.5 x $10^{-8}$ for change in Vm and Rho = 0.45, p = $5.0x10^{-4}$ for change in firing rate) (S7 Fig). Superficial SST neurons located less than 250 μm below the pia on average hyperpolarized by 3.1 ± 2.2 mV (n = 22 cells), which was significantly different from deeper SST neurons located 400–600 μm below the pia which on average depolarised by 0.2 ± 2.2 mV (n = 24 cells) (Tukey-Kramer multiple comparison test, p = 2.3 x $10^{-5}$) (S8 Fig).

## $V_m$ dynamics comparing quiet and whisking states

In some cases, free whisking without object touch continued for more than two seconds after onset, and we compared $V_m$ dynamics during such whisking states with equivalent periods of quiet states, when the whiskers were not moving. We found that excitatory, PV and VIP neurons were on average depolarized during free whisking, but SST neurons were on average hyperpolarized during whisking (Fig 4A). The standard deviation of $V_m$ fluctuations was reduced during whisking for excitatory and PV neurons, but not for VIP or SST neurons (Fig 4B). Comparing AP firing rate, only VIP neurons showed a significant increase during free whisking compared to quiet periods (Fig 4C). Examining grand average FFT spectra of $V_m$ fluctuations, excitatory and PV neurons appeared to show a prominent decrease in slow $V_m$ fluctuations during whisking compared to quiet periods, whereas this was less apparent for VIP and SST neurons (Fig 4D). Nonetheless, when quantified, we found that low frequency (1–10 Hz) $V_m$ fluctuations were significantly suppressed for all cell classes except VIP neurons, whereas high frequency (30–90 Hz) $V_m$ fluctuations were significantly enhanced for all cell classes (Fig 4E). Overall, it seems the excitatory and PV neurons are similarly modulated comparing quiet and whisking states, but VIP neurons as a group uniquely increase firing during whisking, whereas SST neurons as a group uniquely hyperpolarize during whisking. Looking across cell depth, we found that the increase in firing rate during whisking for PV cells was significantly higher for deeper cells and the decrease in standard deviation of $V_m$ was significantly stronger for more superficial PV and excitatory cells (S9–S11 Figs).

## Fast $V_m$ dynamics phase-locked to the whisking cycle

Whisking largely consists of relatively stereotypical backward and forward movements of the whiskers in the frequency range 5–15 Hz, and in our next analyses we computed whisking cycle phase-locked $V_m$ fluctuations (Fig 5A). Excitatory and PV neurons on average showed the largest whisking phase-locked $V_m$ fluctuations, and VIP neurons had the smallest whisking phase-locked $V_m$ fluctuations (Fig 5B). The whisking phase-locked $V_m$ fluctuations were not statistically different for excitatory and PV neurons, but both excitatory and PV neurons had larger whisking phase-locked $V_m$ fluctuations than VIP and SST neurons, which in turn were not significantly different from each other (Fig 5B).

The amplitude of whisking cycle phase-locked $V_m$ fluctuations for the four classes of neurons were: excitatory 1.8 ± 1.2 mV (n = 87 cells); PV 1.5 ± 0.7 mV (n = 43 cells); VIP 0.7 ± 0.4 mV (n = 23 cells); and SST 1.1 ± 0.7 mV (n = 63 cells). Different neurons appeared to have different phase preferences with respect to the whisking cycle (Fig 5C). Excitatory, but not inhibitory, neurons showed stronger phase-locked $V_m$ fluctuations with increased cell depth on average (S12 Fig).

## $V_m$ dynamics evoked by whisker object touch

An object was placed within reach of the C2 whisker in some recording epochs, and during active whisking the mouse palpated the object with repeated whisker-object contacts. Many

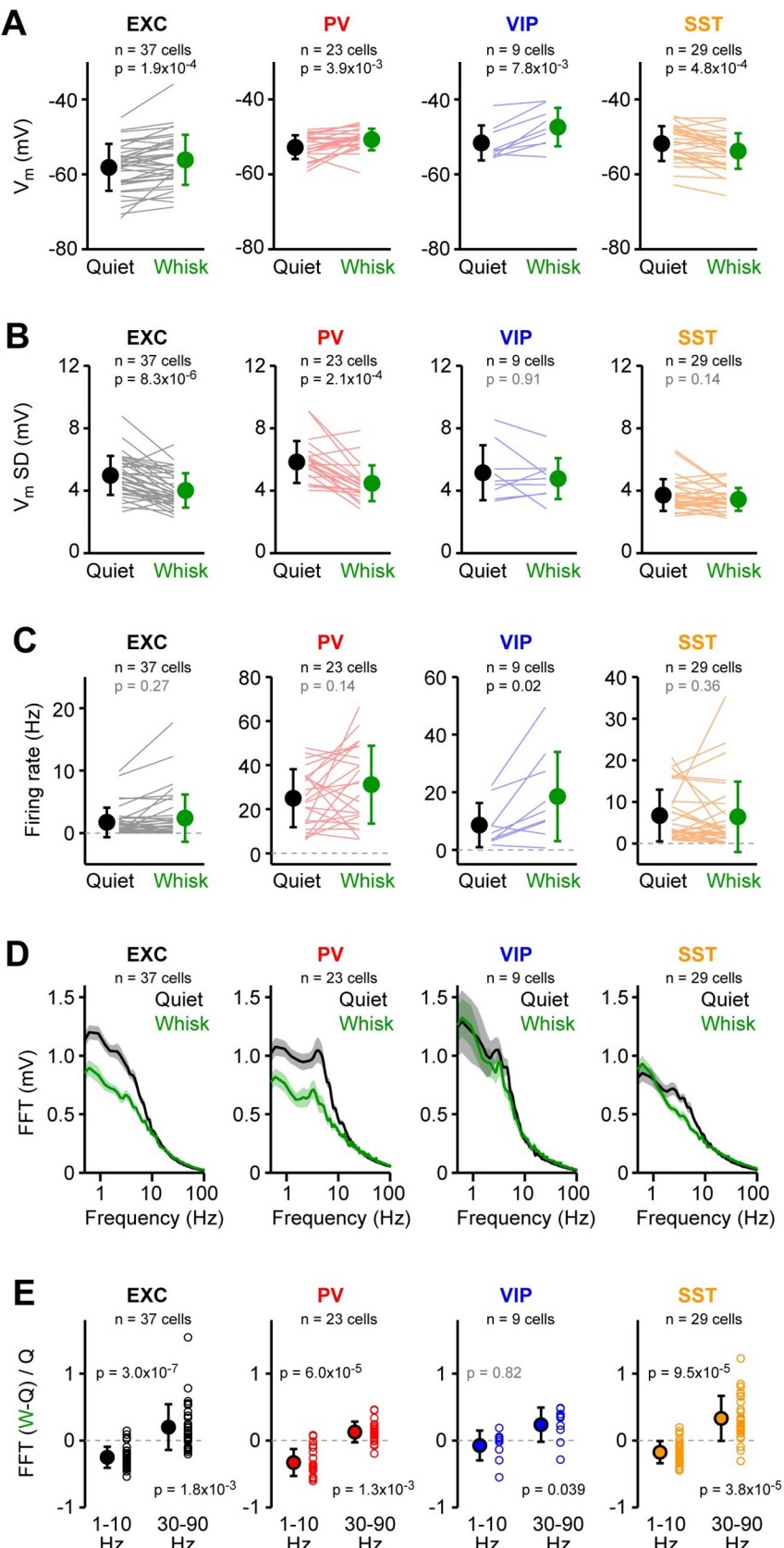

**Fig 4. Membrane potential dynamics across quiet and whisking states.** (**A**) Mean subthreshold $V_m$ computed for 2 s epochs of quiet wakefulness and whisking for the same neurons. Grey lines show individual neurons. Filled circles with error bars show mean ± SD. P values indicate statistical differences (Wilcoxon signed-rank test). (**B**) Same as A, but for the mean standard deviation (SD) of subthreshold $V_m$. (**C**) Same as A, but for the mean firing rate. (**D**) Grand-average FFTs of the subthreshold $V_m$ computed for 2-s epochs of quiet wakefulness and whisking for each cell class. (**E**) Ratio between the mean change in FFT amplitude during whisking relative to quiet wakefulness, (FFTwhisking—FFT$_{quiet}$) / FFT$_{quiet}$, computed for each cell (open circles) in the 1–10 Hz and 30–90 Hz frequency bands. P values indicate statistical differences (Wilcoxon signed-rank test).

neurons responded with rapid changes in $V_m$ upon whisker-object contact (Fig 6A and S13A Fig). Across the four cell classes, we found a large, fast depolarisation upon active whisker touch in excitatory and PV neurons, and somewhat delayed, smaller responses in VIP and SST neurons (Fig 6B, 6C). Across the first 100 ms after whisker-object contact, on average all four groups of cell classes increased AP firing rate, with PV neurons showing a larger increase than excitatory and SST neurons, and VIP neurons having a larger increase than excitatory neurons (Fig 6D). The contact-evoked increase in AP firing rate in the first 100 ms after touch for the

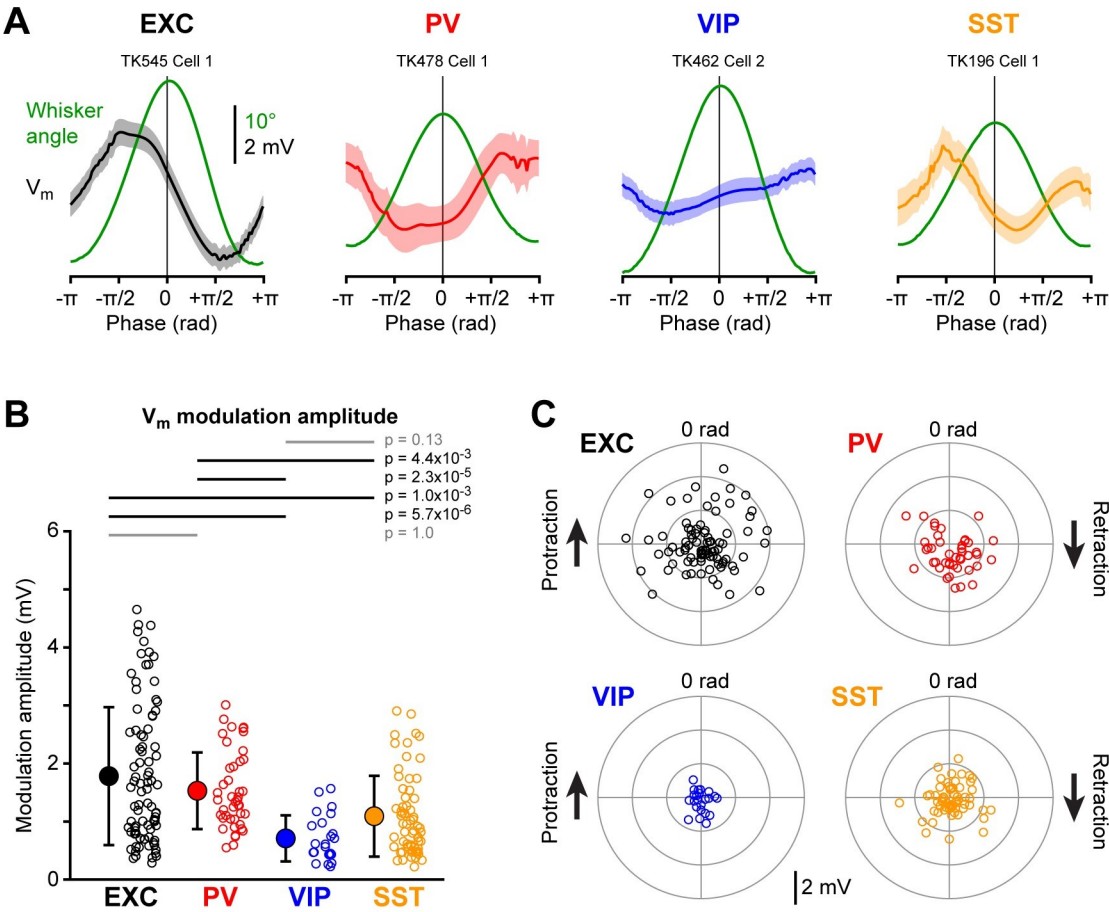

**Fig 5. Fast membrane potential fluctuations phase-locked with the whisking cycle.** (**A**) Example averaged reconstructed angular whisker position (green) and averaged $V_m$ aligned to the phase of the whisking cycle for excitatory (EXC), PV-expressing, VIP-expressing and SST-expressing neurons. (**B**) Mean whisker phase-locked $V_m$ modulation amplitude for the four cell classes. Open circles show individual neuron values. Filled circles with error bars show mean ± SD. Statistical differences between cell classes were computed using a Kruskal-Wallis test (p = 7.8x10$^{-8}$) followed by a Tukey-Kramer multiple comparison test. (**C**) Polar plots showing the mean amplitude of $V_m$ whisking phase-locked modulation vs phase of peak $V_m$ for the four cell classes. Open circles show individual cells.

four classes of neurons were: excitatory 1.6 ± 4.8 Hz (n = 65 cells); PV 41.1 ± 23.4 Hz (n = 21 cells); VIP 18.3 ± 9.4 Hz (n = 8 cells); and SST 10.2 ± 22.3 Hz (n = 36 cells). We computed the change in $V_m$ upon active touch separately for early (5–20 ms) and late (30–100 ms) periods. In the early period, excitatory and PV neurons showed a significant touch-evoked depolarization, but not VIP and SST neurons (Fig 6C). The amplitude of the early depolarization was not significantly different comparing excitatory and PV neurons, but both were larger than the SST group (Fig 6E). In addition, the early touch depolarization in PV neurons was larger than for VIP neurons (Fig 6E). The amplitudes of the early touch-evoked response for the four cell classes were: excitatory 2.1 ± 3.0 mV (n = 65 cells); PV 3.2 ± 2.2 mV (n = 21 cells); VIP 0.2 ± 1.6 mV (n = 8 cells); and SST -0.1 ± 2.1 mV (n = 36 cells). In the late touch-evoked response period, excitatory, PV and SST neurons depolarized significantly, and there were no significant differences comparing among the four cell classes (Fig 6F). The amplitudes of the late touch-evoked response for the four cell classes were: excitatory 2.6 ± 3.1 mV (n = 65 cells); PV 2.0 ± 2.0 mV (n = 21 cells); VIP 2.0 ± 3.2 mV (n = 8 cells); and SST 1.1 ± 2.6 mV (n = 36 cells). Different neurons in each of the 4 cell classes showed variable responses to active contact without strong dependence upon cell depth (S13–S16 Figs).

## Touch evoked $V_m$ dynamics across intercontact intervals

Repeated whisker-object touches can occur at various intercontact intervals (ICIs) giving rise to complex $V_m$ dynamics [34, 35]. Here, we analysed touch-evoked $V_m$ signals according to the time from the previous whisker-object contact. We chose to divide the data into two groups, one with short ICIs of less than 80 ms following the last active touch (mean ICI: 39.3 ± 6.8 ms across n = 131 recordings) and the other group with long ICIs of more than 100 ms since the preceding active touch (mean ICI: 3376.5 ± 2371.2 ms across n = 131 recordings). Individual neurons showed prominent changes in the amplitude of the touch-evoked $V_m$ signals (Fig 7A). Averaged within the four classes of cells, excitatory, PV and VIP neurons appeared to show a strong suppression of the touch-evoked postsynaptic potentials (PSPs) at short ICI (Fig 7B). This depression of responses at short ICI was significant for excitatory and PV neurons (Fig 7C and S17A Fig) and was more pronounced for superficial excitatory and PV cells (S18 and S19 Figs). However, because of temporal integration, the peak depolarization reached in excitatory, PV and VIP neurons during the touch response was very similar for short and long ICI, and we did not find significant modulation by ICI of the peak $V_m$ (Fig 7D and S17B Fig) or evoked firing rate (Fig 7E and S17C Fig) for excitatory, PV and VIP neurons. In contrast, SST neurons had a very different touch-evoked response comparing across short and long ICI. In both individual SST neurons (Fig 7A) and on average across all SST neurons (Fig 7B), SST neurons appeared to change polarity with a small hyperpolarizing touch response at long ICIs reversing to an excitatory PSP at short ICIs (Fig 7B and 7C). This facilitating touch response for SST neurons at high contact frequencies resulted in a significantly depolarised peak $V_m$ at short ICIs (Fig 7D) and a significantly increased AP firing rate at short ICIs (Fig 7E). For short ICIs relative to long ICIs, the peak $V_m$ of SST neurons depolarised by 2.5 ± 2.5 mV (n = 37 cells, p = 4.4x10$^{-6}$) and the firing rate increased by 6.7 ± 11.4 Hz (n = 37 cells, p = 5.2x10$^{-4}$). The facilitation of the response of SST neurons for short ICIs did not appear to depend on the recording depth (S18 and S19 Figs).

## Discussion

Our measurements and analyses extend our knowledge of cell class-dependent and cell depth-dependent $V_m$ dynamics during active whisker sensing. In the following discussion, we highlight our key results for each cell class in the context of previous observations.

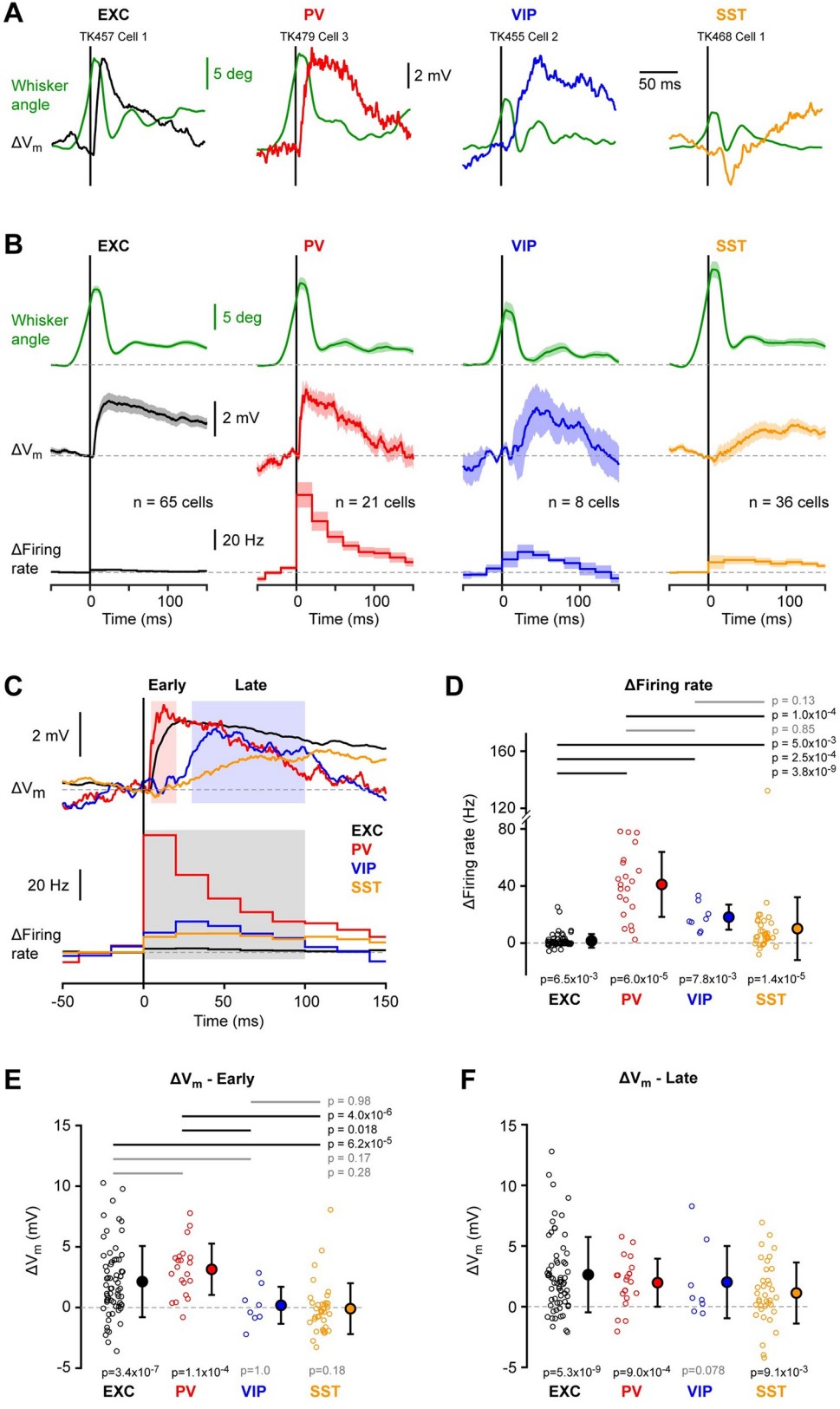

**Fig 6. Active touch-evoked changes in membrane potential.** (**A**) Example angular whisker position (green) and $V_m$ averages at touch onset from excitatory (EXC), PV-expressing, VIP-expressing and SST-expressing neurons. (**B**)

Grand-average whisker angle, change in $V_m$ and change in firing rate at touch onset for each cell class. (**C**) Superimposed grand-average change in $V_m$ (*above*) and in firing rate (*below*) for the four cell classes. The change in $V_m$ was quantified for an early (5–20 ms) and a late (30–100 ms) time-window in panels E and F; the change in firing rate was quantified for a single (0–100 ms) time-window in panel D. (**D**) Mean change in firing rate after touch onset for the four cell classes (0–100 ms time-window). Open circles show individual neuron values. Filled circles with error bars show mean ± SD. Statistical significance for each cell class was tested using a Wilcoxon signed-rank test (p values are shown below); statistical differences between cell classes were computed using a Kruskal-Wallis test (p = 6.5x10$^{-14}$) followed by a Tukey-Kramer multiple comparison test (p values are shown above). (**E**) Same as D, but for the change in $V_m$ in the early (5–20 ms) time-window (Kruskal-Wallis test, p = 4.5x10$^{-7}$). (**F**) Same as E, but for the late (30–100 ms) time-window (Kruskal-Wallis test, p = 0.13; no multiple comparison test).

## Excitatory neurons

The most striking difference between excitatory neurons compared to the various classes of inhibitory neurons was the overall low AP firing rates of the excitatory neurons (Figs 2C and 4C). This is consistent with a large body of literature reporting sparse firing of excitatory neurons in the barrel cortex of awake mice found through extracellular recordings [29, 30, 42, 48, 49], calcium imaging [48, 50–52] and whole cell recordings [29, 30, 34, 35, 42, 44, 49, 53, 54]. The low AP rates likely result from the overall hyperpolarized $V_m$ of excitatory neurons relative to the inhibitory neurons (Figs 2D, 4A and S5 Fig). Excitatory neurons located deeper in the cortex had higher firing rates than superficial neurons (S1 and S2 Figs) and were also more depolarized (S3 and S4 Figs). Interestingly, a similar relationship is found in vitro in brain slices, where neurons in superficial layers are more hyperpolarized than deeper neurons, and the superficial neurons require more current to be injected to initiate AP firing [25].

The excitatory neurons exhibited several interesting features associated with active whisking. Overall, excitatory neurons depolarized at whisking onset, which might be driven by increased excitatory input from the thalamus during whisking [55–57]. Interestingly, shortly after whisking onset, excitatory neurons had a brief transient hyperpolarization (Fig 3B), which is reminiscent of a similar pattern of activity found in superficial neurons of the primary whisker motor cortex wM1 [47]. The mechanisms underlying the transient hyperpolarization need to be further investigated. During prolonged bouts of whisking, the $V_m$ dynamics showed reduced fluctuations in slow frequencies, consistent with previous work indicating reduced slow cortical dynamics during active sensing (Fig 4) [44, 54]. Individual neurons also showed $V_m$ dynamics phase-locked to the whisking cycle, which could contribute to positional coding of whisker-object contacts [58]. Previous work found that these fast phase-locked $V_m$ fluctuations depend upon sensory input from the trigeminal nerve [54] and are primarily relayed via the lemniscal pathway [55]. Consistent with this, given that VPM provides strong input to L3 and L4, but not L2, [24], here we found that deeper-lying neurons, had larger phase-locked $V_m$ fluctuations compared to more superficial neurons (S12A Fig).

Active touch evoked an overall robust depolarization in excitatory neurons with a small increase in firing rate, consistent with previous studies [34, 35, 48] (Fig 6). As revealed by our analysis of the effect of intercontact interval (Fig 7), excitatory neurons tended to show depressing PSP responses during high frequency palpation of an object, but with a more reliable peak $V_m$, in agreement with previous measurements [34]. Such $V_m$ dynamics thus appear to deliver reliable encoding of touch through the peak $V_m$, which also contributes importantly to determining the AP output. Different excitatory cells had different dynamics, which were in part accounted for by cortical depth, with more superficial cells showing stronger suppression at short intercontact intervals, as reported previously [34]. In addition, a previous investigation of excitatory neurons divided according to long-range projection target suggested that neurons in wS1 projecting to wS2 have less frequency-dependent suppression of touch-evoked PSPs compared to neurons in wS1 projecting to wM1 [35].

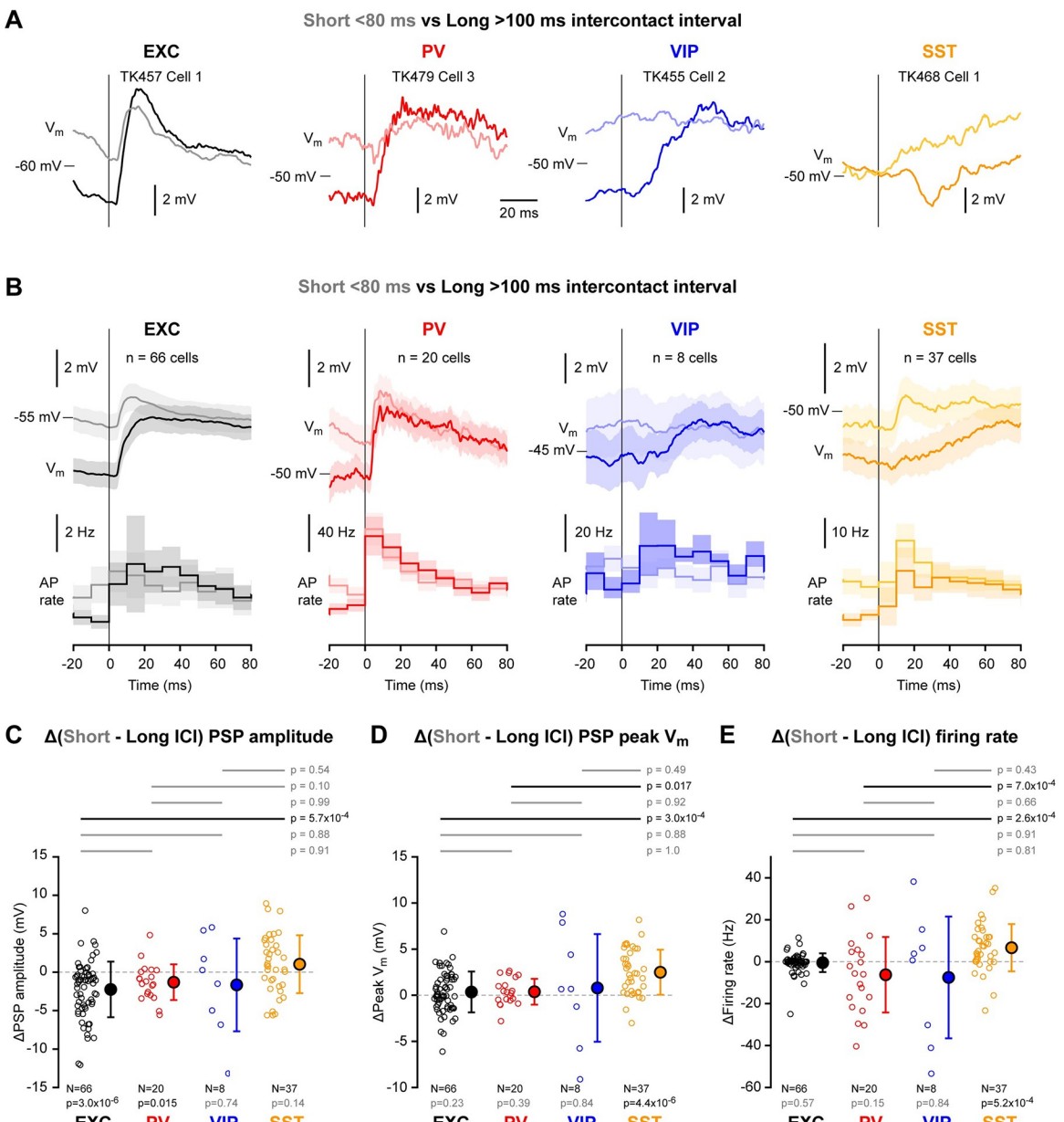

**Fig 7. Frequency-dependent active touch membrane potential signals.** (**A**) Example $V_m$ averages at touch onset for short ($<$ 80 ms, light traces) and long ($>$ 100 ms, dark traces) intercontact intervals from excitatory (EXC), PV-expressing, VIP-expressing and SST-expressing neurons. (**B**) Grand-average $V_m$ and firing rate at touch onset for short and long intercontact-intervals for each cell class. (**C**) Mean difference of PSP amplitude between short and long intercontact intervals for the four cell classes. Open circles show individual neuron values. Filled circles with error bars show mean ± SD. Statistical significance for each cell class was tested using a Wilcoxon signed-rank test (p values are shown below); statistical differences between cell classes were computed using a Kruskal-Wallis test (p = 0.0015) followed by a Tukey-Kramer multiple comparison test (p values are shown above). (**D**) Same as C, but for the mean difference in peak $V_m$ after touch onset (Kruskal-Wallis, p = $4.9 \times 10^{-4}$). (**E**) Same as C, but for the mean difference in firing rate after touch onset (Kruskal-Wallis test, p = $8.9 \times 10^{-5}$).

Overall, our analysis of $V_m$ recordings from excitatory neurons is largely consistent with previous studies and provides an important framework for comparison with the three classes of GABAergic neurons studied here.

## PV-expressing GABAergic neurons

PV cells form the biggest class of GABAergic neurons in the barrel cortex (Fig 1F) and PV neurons fired at higher rates than any of the other neuronal classes (Fig 2C). The high firing rate is likely driven by strong excitatory synaptic input both from nearby excitatory neurons [59–62] and the thalamus [24, 63, 64]. During quiet wakefulness, PV neurons had the largest amplitude slow $V_m$ fluctuations compared to the other cell classes (Fig 2). PV neurons depolarized at whisking onset and increased firing rate (Fig 3), with the increase in AP firing being more prominent in deeper neurons (S7 and S9 Figs), in agreement with a previous study [29]. Similar to excitatory neurons, during prolonged whisking, PV neurons were depolarized with reduced slow $V_m$ fluctuations (Fig 4) and showed prominent whisking phase-locked $V_m$ dynamics (Fig 5). Active touch evoked large and fast depolarization in PV neurons accompanied by a strong increase in AP firing (Fig 6). Similar to excitatory neurons, the touch-evoked responses in PV neurons were strongly suppressed at short intercontact intervals, but the peak $V_m$ was more reliable giving rise to robust coding of touch and AP firing (Fig 7).

Altogether PV neurons had a similar modulation of subthreshold $V_m$ dynamics compared to excitatory neurons. The overall similarity of PV neurons to excitatory neurons might derive from their common strong input from thalamus [24], and the massive local input PV neurons receive from the local excitatory neurons [59]. However, AP firing was much more strongly modulated for PV neurons compared to excitatory neurons, which might derive from their overall more depolarized $V_m$, thus being more likely to cross AP threshold compared to the same input arriving in the more hyperpolarized excitatory neurons. The high firing rates of the PV neurons in turn presumably helps inhibit the excitatory neurons, preventing them from firing at high rates and enforcing sparse coding.

## VIP-expressing GABAergic neurons

VIP neurons differed in many respects from the other classes of neurons. The AP waveform had a duration similar to excitatory neurons and longer than PV and SST neurons (Fig 2B). In recordings from unidentified neurons, it may therefore be difficult to distinguish these neurons from excitatory neurons, although on average they are more depolarised and fire APs at higher rates during quiet wakefulness (Fig 2C, 2D). At the onset of whisking, VIP neurons depolarized and increased firing rate (Fig 3). During prolonged whisking, VIP neurons were depolarized compared to quiet periods and also had increased AP firing rates (Fig 4). The whisking-related depolarization might in part result from nicotinic input to VIP neurons [31] and glutamatergic input from other cortical regions such as wM1 [65, 66], as well as increased thalamic input during whisking, which also provides strong input to a subset of VIP neurons [24]. Although the VIP neurons were prominently excited by whisking, they only exhibited small whisking phase-locked $V_m$ fluctuations (Fig 5). Upon active touch, VIP neurons depolarized and increased AP firing, but this occurred after a substantial delay relative to excitatory and PV neurons (Fig 6). The late excitation of VIP neurons during active touch is consistent with their late excitation in response to passively applied whisker deflections in a detection task [67]. In agreement with a previous study [29], VIP neurons therefore appear to be less directly affected by whisker sensory input than excitatory or PV neurons. Presumably, this reflects, at least in part, the overall weak primary sensory VPM thalamic input to the majority of VIP neurons [24]. VIP neurons might therefore be most prominently regulated by the overall behavioral state (in our experiments whisking vs quiet), rather than specific aspects of whisker sensory processing. This would be consistent with the view that VIP neurons might serve an overall disinhibitory function [65, 68, 69], perhaps primarily gating the impact of attention, context and motor-related inputs to barrel cortex.

## SST-expressing GABAergic neurons

SST neurons also differed strongly from the other classes of barrel cortex neurons we examined in this study. Consistent with previous measurements in barrel cortex [32], SST neurons had significantly smaller amplitude slow $V_m$ fluctuations than the three other classes of neurons (Fig 2F, 2G). On average, SST neurons hyperpolarized at the onset of whisking and continued to remain hyperpolarized during prolonged whisking, unlike the other three classes of neurons, which depolarized (Figs 3 and 4) [32, 65]. The hyperpolarization of SST neurons has been suggested to result from the excitation of VIP neurons, which are known to strongly innervate SST neurons [65, 68], giving rise to the widely-accepted overall disinhibitory function of VIP neurons [65]. Interestingly, the hyperpolarization at whisking onset was prominent for superficial SST neurons, but not for neurons deeper in the cortex (Fig 3E). This result is consistent with the finding that whisking-related muscarinic input depolarizes a non-Martinotti subtype of SST neuron found most prominently in deeper cortical layers [36].

Many SST neurons showed weak phase-locked $V_m$ dynamics, on average significantly smaller than for excitatory and PV neurons (Fig 5). The average active touch response was small and delayed relative to the fast and large touch responses in excitatory and PV neurons. These relatively weak active touch responses in SST neurons are consistent with the near-absent AP responses evoked by passive whisker stimulation in a detection task [67]. Fast sensory responses are presumably, at least in part, driven by direct thalamic input to barrel cortex, and in vitro measurements have shown little synaptic input from VPM and POm onto SST neurons [24, 70]. Interestingly, SST neurons showed a unique relationship to intercontact interval compared to excitatory, PV and VIP neurons. At long intercontact intervals, SST neurons on average showed a small hyperpolarization, but at short intercontact intervals, this turned into an excitatory PSP response (Fig 7). The biophysical properties of synaptic transmission from local presynaptic excitatory neurons onto postsynaptic SST neurons may account for this change in touch-evoked responses in SST neurons. Both in vitro [71, 72] and in vivo [62], excitatory synaptic input to SST neurons shows prominent short-term facilitation. Thus, during repetitive active touch at high frequencies, the same excitatory neurons in cortex could fire repeatedly giving rise to facilitating synaptic input onto SST neurons, which in our analysis would be revealed as increased depolarizing responses at short intercontact intervals.

## Future perspectives

Our measurements and analyses provide a more complete view of $V_m$ dynamics in barrel cortex during active whisker sensing, highlighting important differences between excitatory, PV, VIP and SST classes of neurons in the upper half (<600 μm below the pia) of the cortical thickness. In further analyses of the dataset, it could be interesting to train classifiers based on our cell class-labelled data, which might then be useful to help identify cells recorded under other conditions where no cell class labels are present. In the future, it will also be of great importance to investigate $V_m$ of neurons in the deeper layers 5 and 6. However, targeting whole-cell recordings through two-photon imaging to such deep neurons is very challenging, and might require the application of new technologies such as three-photon imaging, which promises increased contrast for deep tissue imaging [73].

Another important limitation of the current dataset, is that we have not distinguished between different subtypes of neurons within the four classes we examined. For example, previous studies have differentiated excitatory neurons projecting to wM1 and wS2, which were found to have different $V_m$ dynamics during active sensing [35]. Indeed, even within layer 2/3, there are excitatory neurons with many other projection targets [74], and they may exhibit different activity patterns during active whisker sensation. Equally, it is increasingly clear that

there are many subclasses of PV, VIP and SST neurons based both on transcriptomic analyses, anatomical features and connectivity, as well as additional groups of GABAergic neurons not expressing PV, VIP or SST [14, 15, 36, 75]. Through intersectional strategies, for example combining various Cre and Flp lines [76], it is becoming possible to specifically target distinct subsets of the relatively broad classes we investigated in this study. However, because of the complexity of breeding and studying many mouse lines, it will probably take some time before detailed analyses of the $V_m$ dynamics of the various subclasses are investigated in quantitative comparative studies. Indeed, as the number of different subclasses of neurons increases, the approach of studying these through whole-cell recordings becomes less feasible due to the technical difficulty in obtaining these recordings. Fortunately, the signal-to-noise ratio of fluorescent $V_m$ sensors for in vivo imaging with cellular resolution is rapidly increasing [77], and this methodology will likely be necessary to examine many subclasses of GABAergic and excitatory neurons in combination with fluorescence in situ hybridisation (FISH) to molecularly characterize cell types [78].

It will also be important to examine the various types of barrel cortex neurons during different behaviors, for example whisker detection tasks, object localization tasks, shape discrimination or texture discrimination tasks. It is quite possible that different neuron types will play different roles depending upon behavioral demands. Perhaps of particular interest, would be to investigate whether disinhibitory mechanisms might be necessary for top-down context-dependent processing and reward-based learning.

Finally, the complexity of neuronal circuit function demands detailed computational modelling to further mechanistic understanding, which becomes increasingly realistic in view of increased biological knowledge along with enhanced computational power [28, 79]. In particular, connectomic approaches through electron microscopy and dense reconstruction to delineate cortical connectivity [80, 81] will likely provide essential information for computational network modelling, which could then be compared to in vivo $V_m$ dynamics coupled with optogenetic manipulations for causal mechanistic understanding. There is therefore much work ahead before we can reach a complete understanding of the functioning of a barrel cortex column.

## Supporting information

**S1 Video. Spatial distribution of Scnn1a-expressing neurons in the C2 barrel column.** Related to Fig 1. Compressed and downsampled stack of confocal images of tdTomato fluorescence in the C2 barrel column of a Scnn1a-Cre x LSL-tdTomato mouse after fixation and CUBIC tissue clearing.
(MOV)

**S2 Video. Spatial distribution of PV-expressing neurons in the C2 barrel column.** Related to Fig 1. Compressed and downsampled stack of confocal images of tdTomato fluorescence in the C2 barrel column of a PV-Cre x LSL-tdTomato mouse after fixation and CUBIC tissue clearing.
(MOV)

**S3 Video. Spatial distribution of VIP-expressing neurons in the C2 barrel column.** Related to Fig 1. Compressed and downsampled stack of confocal images of tdTomato fluorescence in the C2 barrel column of a VIP-Cre x LSL-tdTomato mouse after fixation and CUBIC tissue clearing.
(MOV)

**S4 Video. Spatial distribution of SST-expressing neurons in the C2 barrel column.** Related to Fig 1. Compressed and downsampled stack of confocal images of tdTomato fluorescence in the C2 barrel column of a SST-Cre x LSL-tdTomato mouse after fixation and CUBIC tissue clearing.
(MOV)

**S1 Fig. Action potential firing across cortical depth during quiet wakefulness.** Related to Fig 2. (**A**) Mean action potential (AP) duration for epochs of quiet wakefulness across cell depth for each cell class. Open circles represent single neurons. Correlation between firing rate and cell depth was assessed using a Spearman test; Spearman correlation coefficient (Rho) and p value are indicated on each graph. (**B**) Same as A, but for the mean AP firing rate. (**C**) Same as A, but for the mean AP threshold.
(JPG)

**S2 Fig. Action potential firing across cortical layers during quiet wakefulness.** Related to Fig 2. (**A**) Mean action potential (AP) duration for epochs of quiet wakefulness across cortical layers for each cell class. Bars and error bars represent mean and SD, respectively. The number of cells in each layer is indicated below each bar. Statistical differences between layers were tested using a Kruskal-Wallis test (EXC, $p = 1.8 \times 10^{-4}$; PV, $p = 0.29$; VIP, $p = 0.18$; SST, $p = 0.12$) followed by a Tukey-Kramer multiple comparison test, when appropriate (p values indicated on the graph in grey or black for non-significant and significant differences, respectively). (**B**) Same as A, but for the mean AP firing rate (Kruskal-Wallis: EXC, $p = 9.0 \times 10^{-7}$; PV, $p = 0.014$; VIP, $p = 0.012$; SST, $p = 0.55$). (**C**) Same as A, but for the mean AP threshold (Kruskal-Wallis: EXC, $p = 0.17$; PV, $p = 0.13$; VIP, $p = 0.62$; SST, $p = 0.018$).
(JPG)

**S3 Fig. Membrane potential fluctuations across cortical depth during quiet wakefulness.** Related to Fig 2. (**A**) Mean membrane potential ($V_m$) for epochs of quiet wakefulness across cell depth for each cell class. Open circles represent single neurons. Correlation between mean $V_m$ and cell depth was assessed using Spearman test; Spearman correlation coefficient (Rho) and p value are indicated on each graph. (**B**) Same as A, but for the mean standard deviation (SD) of the $V_m$. (**C**) Same as A, but for the mean 1–10 Hz $V_m$ FFT amplitude.
(JPG)

**S4 Fig. Membrane potential fluctuations across cortical layers during quiet wakefulness.** Related to Fig 2. (**A**) Mean membrane potential ($V_m$) for epochs of quiet wakefulness across cortical layers for each cell class. Bars and error bars represent mean and SD, respectively. The number of cells in each layer is indicated below each bar. Statistical differences between layers were tested using a Kruskal-Wallis test (EXC, $p = 0.016$; PV, $p = 0.11$; VIP, $p = 0.40$; SST, $p = 0.83$) followed by a Tukey-Kramer multiple comparison test, when appropriate (p values indicated on the graph in grey or black for non-significant and significant differences, respectively). (**B**) Same as A, but for the mean standard deviation (SD) of the $V_m$ (Kruskal-Wallis: EXC, $p = 0.23$; PV, $p = 3.7 \times 10^{-4}$; VIP, $p = 0.50$; SST, $p = 1.4 \times 10^{-4}$). (**C**) Same as A, but for the mean 1–10 Hz $V_m$ FFT amplitude (Kruskal-Wallis: EXC, $p = 0.34$; PV, $p = 9.0 \times 10^{-4}$; VIP, $p = 0.21$; SST, $p = 7.2 \times 10^{-4}$).
(JPG)

**S5 Fig. Action potential firing versus membrane potential during quiet wakefulness.** Related to Fig 2. (**A**) Mean firing rate (logarithmic scale) vs mean AP duration. Open triangles represent single excitatory neurons. Open circles represent inhibitory neurons expressing PV (red), VIP (blue) or SST (orange). Correlation between $\text{Log}_{10}$(firing rate) and AP duration was

assessed using a Pearson test. (**B**) Same as A, but for the mean firing rate vs AP threshold. (**C**) Same as A, but for the mean firing rate vs mean $V_m$. (**D**) Same as A, but for the mean firing rate vs mean standard deviation (SD) of the $V_m$.
(JPG)

**S6 Fig. Membrane potential dynamics preceding action potential initiation during quiet wakefulness.** Related to Fig 2. (**A**) Grand-average spike-triggered $V_m$ for excitatory (EXC, black) and inhibitory neurons expressing PV (red), VIP (blue) or SST (orange). The grey dashed-line indicates the mean AP threshold across all cell classes. (**B**) Depolarization before AP initiation computed as the change in $V_m$ 20 ms before AP initiation relative to AP threshold. Open circles show individual neuron values. Filled circles with error bars show mean ± SD. Statistical differences between cell classes were computed using a Kruskal-Wallis test (p = $1.5 \times 10^{-11}$) followed by a Tukey-Kramer multiple comparison test. (**C**) Mean AP threshold. Open circles show individual neuron values. Filled circles with error bars show mean ± SD. Statistical differences between cell classes were computed using a Kruskal-Wallis test (p = $7.2 \times 10^{-4}$) followed by a Tukey-Kramer multiple comparison test. (**D**) Same as A, but computed only for all APs not preceded by another AP within 25 ms. (**E**) Same as B, but computed for APs with 25 ms exclusion window before AP (Kruskal-Wallis test, p = $2.1 \times 10^{-14}$). (**F**) Same as C, but computed for APs with 25 ms exclusion window before AP (Kruskal-Wallis test, p = $8.7 \times 10^{-4}$).
(JPG)

**S7 Fig. Membrane potential changes at the onset of whisking across cortical depth.** Related to Fig 3. (**A**) Change in $V_m$ at whisking onset across cell depth for each cell class. Open circles represent single neurons. Correlation between $V_m$ change and cell depth was assessed using a Spearman test; Spearman correlation coefficient (Rho) and p value are indicated on each graph. (**B**) Same as A, but for the change in firing rate. (**C**) Change in firing rate vs change in $V_m$ at whisking onset for the four cell classes. Correlation between the change in firing rate and the change in $V_m$ was assessed using a Pearson test; Pearson correlation coefficient (r) and p value are indicated on each graph.
(JPG)

**S8 Fig. Membrane potential changes at the onset of whisking across cortical layers.** Related to Fig 3. (**A**) Change in $V_m$ at whisking onset across cortical layers for each cell class. Bars and error bars represent mean and SD, respectively. The number of cells in each layer is indicated below each bar. Statistical differences between layers were tested using a Kruskal-Wallis test (EXC, p = 0.74; PV, p = 0.30; SST, p = $4.5 \times 10^{-5}$) followed by a Tukey-Kramer multiple comparison test, when appropriate (p values indicated on the graph in grey or black for non-significant and significant differences, respectively). (**B**) Same as A, but for the change in firing rate (Kruskal-Wallis: EXC, p = 0.61; PV, p = 0.14; SST, p = 0.012).
(JPG)

**S9 Fig. Changes in membrane potential dynamics in quiet vs whisking states across cortical depth.** Related to Fig 4. (**A**) Difference in firing rate (whisking minus quiet wakefulness) across cell depth for each cell class. Open circles represent single neurons. Correlation between difference in firing rate and cell depth was assessed using a Spearman test; Spearman correlation coefficient (Rho) and p value are indicated on each graph. (**B**) Same as A, but for the difference in mean $V_m$. (**C**) Same as A, but for the difference in mean standard deviation (SD) of the $V_m$. (**D**) Same as A, but for the relative change in 1–10 Hz $V_m$ FFT amplitude (whisking minus quiet wakefulness divided by quiet wakefulness). (**E**) Same as A, but for the relative change in

30–90 Hz $V_m$ FFT amplitude.
(JPG)

**S10 Fig. Changes in membrane potential dynamics in quiet vs whisking states across cortical layers.** Related to Fig 4. (**A**) Difference in firing rate (whisking minus quiet wakefulness) across cortical layers for each cell class. Bars and error bars represent mean and SD, respectively. The number of cells in each layer is indicated below each bar. Statistical differences between layers were tested using a Kruskal-Wallis test (EXC, p = 0.55; PV, p = 0.27; SST, p = 0.15) followed by a Tukey-Kramer multiple comparison test, when appropriate (p values indicated on the graph in grey or black for non-significant and significant differences, respectively). (**B**) Same as A, but for the difference in mean $V_m$ (Kruskal-Wallis test, EXC, p = 0.36; PV, p = 0.49; SST, p = 0.090). (**C**) Same as A, but for the difference in mean standard deviation (SD) of the $V_m$ (Kruskal-Wallis test, EXC, p = 0.030; PV, p = 0.046; SST, p = 0.65).
(JPG)

**S11 Fig. Changes in membrane potential FFT amplitude in quiet vs whisking states across cortical layers.** Related to Fig 4. (**A**) Relative change in mean 1–10 Hz FFT amplitude (whisking minus quiet wakefulness divided by quiet wakefulness) across cortical layers for each cell class. Bars and error bars represent mean and SD, respectively. The number of cells in each layer is indicated below each bar. Statistical differences between layers were tested using a Kruskal-Wallis test (EXC, p = 0.037; PV, p = 0.075; SST, p = 0.94) followed by a Tukey-Kramer multiple comparison test, when appropriate (p values indicated on the graph in grey or black for non-significant and significant differences, respectively). (**B**) Same as A, but for the difference in 30–90 Hz $V_m$ FFT amplitude (Kruskal-Wallis test: EXC, p = 0.24; PV, p = 0.13; SST, p = 0.66).
(JPG)

**S12 Fig. Fast membrane potential fluctuations phase-locked with the whisking cycle across cortical depth.** Related to Fig 5. (**A**) Mean $V_m$ phase modulation amplitude across cell depth for each cell class. Open circles represent single neurons. Correlation between the modulation amplitude and cell depth was assessed using a Spearman test; Spearman correlation coefficient (Rho) and p value are indicated on each graph. (**B**) Mean $V_m$ phase modulation amplitude across cortical layers for each cell class. Bars and error bars represent mean and SD, respectively. The number of cells in each layer is indicated below each bar. Statistical differences between layers were tested using a Kruskal-Wallis test (EXC, p = 0.041; PV, p = 0.35; VIP, p = 0.19; SST, p = 0.30) followed by a Tukey-Kramer multiple comparison test, when appropriate (p values indicated on the graph in grey or black for non-significant and significant differences, respectively). (**C**) Same as A, but for the phase preference.
(JPG)

**S13 Fig. Active touch-evoked changes in subthreshold membrane potential across cortical depth.** Related to Fig 6. (**A**) Color-coded average single-neuron $V_m$ responses to active touch onset for each cell class. Neurons are sorted by the amplitude of the peak response. (**B**) Change in $V_m$ in the early (5–20 ms) time window after touch onset across cell depth for each cell class. Open circles represent single neurons. Correlation between $V_m$ change and cell depth was assessed using a Spearman test; Spearman correlation coefficient (Rho) and p value are indicated on each graph. (**C**) Same as B, but for the late (30–100 ms) time window after touch onset.
(JPG)

**S14 Fig. Active touch-evoked changes in subthreshold membrane potential across cortical layers.** Related to Fig 6. (**A**) Change in $V_m$ in the early (5–20 ms) time window after touch onset across cortical layers for each cell class. Bars and error bars represent mean and SD, respectively. The number of cells in each layer is indicated below each bar. Statistical differences between layers were tested using a Kruskal-Wallis test (EXC, p = 0.38; PV, p = 0.26; SST, p = 0.58) followed by a Tukey-Kramer multiple comparison test, when appropriate (p values indicated on the graph in grey or black for non-significant and significant differences, respectively). (**B**) Same as B, but for the late (30–100 ms) time window after touch onset (Kruskal-Wallis test: EXC, p = 0.011; PV, p = 0.058; SST, p = 0.89).
(JPG)

**S15 Fig. Active touch-evoked changes in sub- and supra-threshold membrane potential across cortical depth.** Related to Fig 6. (**A**) Change in $V_m$ (0–100 ms) after touch onset across cell depth for each cell class. Open circles represent single neurons. Correlation between $V_m$ change and cell depth was assessed using a Spearman test; Spearman correlation coefficient (Rho) and p value are indicated on each graph. (**B**) Same as A, but for the change in firing rate. (**C**) Change in firing rate vs change in $V_m$. The correlation was assessed using a Pearson test; Pearson correlation coefficient (r) and p value are indicated on each graph.
(JPG)

**S16 Fig. Active touch-evoked changes in sub- and supra-threshold membrane potential across cortical layers.** Related to Fig 6. (**A**) Change in $V_m$ (0–100 ms) after touch onset across cortical layers for each cell class. Bars and error bars represent mean and SD, respectively. The number of cells in each layer is indicated below each bar. Statistical differences between layers were tested using a Kruskal-Wallis test (EXC, p = 0.014; PV, p = 0.12; SST, p = 0.91) followed by a Tukey-Kramer multiple comparison test, when appropriate (p values indicated on the graph in grey or black for non-significant and significant differences, respectively). (**B**) Same as A, but for the change in firing rate (Kruskal-Wallis: EXC, p = 0.85; PV, p = 0.23; SST, p = 0.75).
(JPG)

**S17 Fig. Active touch-evoked changes in membrane potential for short and long intercontact intervals.** Related to Fig 7. (**A**) Postsynaptic potential (PSP) amplitude for short vs long intercontact intervals for each cell class. Open circles represent single neurons. (**B**) Same as A, but for peak PSP $V_m$. (**C**) Same as A, but for firing rate. (**D**) Difference in firing rate vs difference in PSP amplitude between short and long intercontact intervals. Correlation was assessed using a Pearson test; Pearson correlation coefficient (r) and p value are indicated on each graph. (**E**) Same as D, but for the difference in firing rate vs the difference in peak PSP $V_m$.
(JPG)

**S18 Fig. Active touch-evoked changes in membrane potential for short and long intercontact intervals across cortical depth.** Related to Fig 7. (**A**) Mean difference in postsynaptic potential (PSP) amplitude between short and long intercontact intervals across cell depth for each cell class. Open circles represent single neurons. Correlation between the difference in PSP amplitude and cell depth was assessed using a Spearman test; Spearman correlation coefficient (Rho) and p value are indicated on each graph. (**B**) Same as A, but for the difference in $V_m$ at the peak of the PSP. (**C**) Same as A, but for the difference in firing rate.
(JPG)

**S19 Fig. Active touch-evoked changes in membrane potential for short and long intercontact intervals across cortical layers.** Related to Fig 7. (**A**) Mean difference in postsynaptic

potential (PSP) amplitude between short and long intercontact intervals across cortical layers for each cell class. Bars and error bars represent mean and SD, respectively. The number of cells in each layer is indicated below each bar. Statistical differences between layers were tested using a Kruskal-Wallis test (EXC, p = 0.015; PV, p = 0.075; SST, p = 0.97) followed by a Tukey-Kramer multiple comparison test when appropriate (p values indicated on the graph in grey or black for non-significant and significant differences, respectively). (**B**) Same as A, but for the difference in $V_m$ at the peak of the PSP (Kruskal-Wallis test: EXC, p = 0.94; PV, p = 0.59; SST, p = 0.59). (**C**) Same as A, but for the difference in firing rate (Kruskal-Wallis test: EXC, p = 0.58; PV, p = 0.38; SST, p = 0.26).
(JPG)

## Acknowledgments

The authors thank Takayuki Yamashita and other members of the Petersen laboratory for helpful scientific and technical discussions.

## Author Contributions

**Conceptualization:** Taro Kiritani, Sylvain Crochet, Carl C. H. Petersen.

**Data curation:** Taro Kiritani, Aurélie Pala, Célia Gasselin, Sylvain Crochet.

**Formal analysis:** Taro Kiritani, Sylvain Crochet.

**Funding acquisition:** Carl C. H. Petersen.

**Investigation:** Taro Kiritani, Aurélie Pala, Célia Gasselin.

**Methodology:** Taro Kiritani, Aurélie Pala, Célia Gasselin.

**Software:** Sylvain Crochet.

**Supervision:** Sylvain Crochet, Carl C. H. Petersen.

**Visualization:** Sylvain Crochet, Carl C. H. Petersen.

**Writing – original draft:** Sylvain Crochet, Carl C. H. Petersen.

**Writing – review & editing:** Taro Kiritani, Aurélie Pala, Célia Gasselin, Sylvain Crochet, Carl C. H. Petersen.

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
