## [Decision Letter · Decision Letter 0]

7 Feb 2023

PONE-D-22-33584Membrane potential dynamics of excitatory and inhibitory neurons in mouse barrel cortex during active whisker sensingPLOS ONE

Dear Dr. Petersen,

Thank you for submitting your manuscript to PLOS ONE. After careful consideration, we feel that it has merit but does not fully meet PLOS ONE’s publication criteria as it currently stands. Therefore, we invite you to submit a revised version of the manuscript that addresses the points raised during the review process.

Both reviewers and I agree that you have provided a very nice study, and we would like to congratulate your and your co-workers on this. Both reviewers have a number of suggestions to improve the presentation of the work, and to clarify some aspects. I do hope you will find these remarks useful when drafting an updated version.

We look forward to receiving your revised manuscript.

Kind regards,

Laurens W. J. Bosman, Ph.D.

Academic Editor

PLOS ONE

Journal Requirements:

When submitting your revision, we need you to address these additional requirements. 1. Please ensure that your manuscript meets PLOS ONE's style requirements, including those for file naming. The PLOS ONE style templates can be found at https://journals.plos.org/plosone/s/file?id=wjVg/PLOSOne_formatting_sample_main_body.pdf and https://journals.plos.org/plosone/s/file?id=ba62/PLOSOne_formatting_sample_title_authors_affiliations.pdf 2. Please update your submission to use the PLOS LaTeX template. The template and more information on our requirements for LaTeX submissions can be found at http://journals.plos.org/plosone/s/latex. 3. Thank you for stating in your Funding Statement: "This work was supported by project grant 310030_146252 from Swiss National Science Foundation (https://www.snf.ch/en) (CCHP) and Advanced grant 293660 from the European Research Council (https://erc.europa.eu) (CCHP)." Please provide an amended statement that declares *all* the funding or sources of support (whether external or internal to your organization) received during this study, as detailed online in our guide for authors at http://journals.plos.org/plosone/s/submit-now.  Please also include the statement “There was no additional external funding received for this study.” in your updated Funding Statement. Please include your amended Funding Statement within your cover letter. We will change the online submission form on your behalf. 4. We note that you have stated that you will provide repository information for your data at acceptance. Should your manuscript be accepted for publication, we will hold it until you provide the relevant accession numbers or DOIs necessary to access your data. If you wish to make changes to your Data Availability statement, please describe these changes in your cover letter and we will update your Data Availability statement to reflect the information you provide. 5. Please include your full ethics statement in the ‘Methods’ section of your manuscript file. In your statement, please include the full name of the IRB or ethics committee who approved or waived your study, as well as whether or not you obtained informed written or verbal consent. If consent was waived for your study, please include this information in your statement as well.  6. Please review your reference list to ensure that it is complete and correct. If you have cited papers that have been retracted, please include the rationale for doing so in the manuscript text, or remove these references and replace them with relevant current references. Any changes to the reference list should be mentioned in the rebuttal letter that accompanies your revised manuscript. If you need to cite a retracted article, indicate the article’s retracted status in the References list and also include a citation and full reference for the retraction notice.

Reviewers' comments:

Reviewer's Responses to Questions

**Comments to the Author**

1. Is the manuscript technically sound, and do the data support the conclusions?

Reviewer #1: Yes

Reviewer #2: Yes

2. Has the statistical analysis been performed appropriately and rigorously? 

Reviewer #1: Yes

Reviewer #2: Yes

3. Have the authors made all data underlying the findings in their manuscript fully available?

Reviewer #1: Yes

Reviewer #2: No

4. Is the manuscript presented in an intelligible fashion and written in standard English?

Reviewer #1: Yes

Reviewer #2: Yes

5. Review Comments to the Author

Reviewer #1: In this manuscript, Kiritani et al. describe the membrane potential dynamics of multiple classes of excitatory and inhibitory cells across cortical layers in the primary whisker somatosensory barrel cortex (wS1) of awake head-fixed mice during various behavioral conditions (quiet wakefulness, free whisking, and active touch). To this aim, they used whole-cell two-photon guided recordings targeted to fluorescently-labelled neurons in the C2 barrel column while monitoring whisker movements with a camera. They found that the four different classes of recorded neurons had different supra-threshold and sub-threshold dynamics of their membrane potential during quiet wakefulness. Moreover, PV, VIP, and excitatory cells depolarized upon whisking, while SST interneurons hyperpolarized at the onset of whisker movements. The hyperpolarization of the membrane potential of SST cells at the beginning of whisking was a decreasing function of cell depth, being almost absent for deeper SST cells. Upon active whisker touch, excitatory and PV neurons showed pronounced depolarization, while VIP and SST neurons displayed smaller and delayed responses to touch. The author finally analyzed cellular responses to touch as a function of the inter-contact interval (ICI). While excitatory, PV, and VIP cells displayed suppression of the touch-evoked PSP at short ICIs, SST neurons switched response polarity as a function of the ICI (hyperpolarizing at long ICIs and depolarizing at short ICIs).

This study provides a systematic and well-conceived characterization of the supra- and sub-threshold response of different cell types within wS1 in awake head-fixed mice during different behavioral conditions. Results are well presented, the text is clear, and the discussion addresses a sufficient number of critical issues. I support publication of the current version of the manuscript in PLOS One. I only have a couple of very minor suggestions (please find them below).

1) Page 9, line 17 from top, “Many neurons responded with rapid changes…”: please include reference to Figure 6 Figure Supplement 1, so that the reader may quantitatively evaluate how many neurons respond.

1) Page 16, lines 10-12 from bottom, “.. it is increasingly clear that there are many subclasses of PV, VIP and SST neurons based both on transcriptomic analyses, anatomical features and connectivity”: please include appropriate reference for the transcriptomic, anatomical, and connectivity classification.

Reviewer #2: Kiritani et al. use whole-cell patch clamp recordings from both excitatory and inhibitory neurons across the cortical depth to characterize cell-type specific responses to quiet states, free whisking, and active whisker touch. Although prior studies have investigated this using extracellular recordings, the current study expands this work by recording both sub- and suprathreshold responses in both superficial and deep layers using two-photon guided whole-cell recordings to record from excitatory, PV, SST, and VIP neurons. Much of the data had been collected as part of prior studies but reanalyzed here for direct comparisons, and this will be a useful reference in the field. The data provides a good foundation for further investigation into circuit changes driven by other behaviors such as detection or discrimination learning. A major strength of this manuscript is the ability to compare across different brains states which is important for furthering our understanding of sensory processing. However, the authors have not fully taken advantage of their ability to target neurons at multiple depths to compare cell-type specific responses in different layers, except in some supplemental figures which are not well-integrated into the text.

Major Concerns:

1. The authors justify the need for whole-cell recordings in vivo, but then do not use their data to make any mechanistic inferences. In Figure 3B, the brief hyperpolarization of EXT (and also, it appears, PV) at whisker onset could be aligned to the activity of a specific class of interneuron. What is the most likely candidate? If it is PV neurons, what might this mean? Another exciting possibility enabled by this investigation is that the authors might be able to leverage this labeled dataset to build a classifier for future recording and cell identification. Was this attempted? This opportunity should be acknowledged in the discussion at least.

2. The authors made claims about how “previous extracellular measurements of cell class-specific AP firing in barrel cortex has revealed prominent differences depending upon the depth of the cell body relative to the pial surface (Muñoz et al., 2017; Yu et al., 2019, 2016)” Yet, for the main figures the only data they split and analyzed by depth was SST (Figure 3E). Were there prominent differences in PV responses between layers? Supplemental figures did convey data based on depth but the way it is presented was complicated and sometimes not easy to follow.

3. The analysis is typically focused on very short time periods before and after whisking onset. Can the authors create a histogram with a longer time period of analysis? This may be useful for investigators using stimulation paradigms where responses are not limited to 100 ms, such as learning tasks or delayed match-to-sample tasks.

4. The trimming of all whiskers but C2, and the use of a single whisker stimulus is highly artificial. Did the authors have recordings where animals were using the entire whisker apparatus? Did this change the response properties of their identified neurons? How long before the recording session was trimming carried out? Were there changes in the response properties of neurons due to trimming?

Minor Concerns:

1. The authors should indicate what state they analyzed to extract mean firing rates. Were animals locomoting during quiet wakefulness?

2. The authors described in the introduction that a limitation in the current Vm studies is the depth at which they can achieve. Yet, based on what is described in the text it suggests this study is still focused on superficial layers. While the depth correlation analysis found in several supplemental figures was informative, it was not straightforward to read or interpret. Grouping cell properties based on lamination and comparing values would provide a more comprehensive interpretation of some major point being conveyed. For example, in Fig 6 Sup 1 comparing the early vs late responses for L2/3, L4 and L5 cells would provide a more comprehensible view of the data being presented.

3. What is the difference between Fig 1 E and F? There appears to be a discrepancy in number of cells recorded based on depth. Also why is there no reconstruction of EXC?

4. The authors report they can achieve depths up to 600um but Figure 1F shows reconstructed cells in 800-1200um depths.

5. The statistics and cell numbers were not always easy to read on the graphs. The gray for non-significant samples was also very difficult to read.

6. “Consistent with this, given that VPM provides strong input to L3 and L4 (Sermet et al., 2019), here we found that deeper lying neurons in layers 3 and 4, had larger phase-locked Vm fluctuations compared to more superficial neurons.” It is surprising that given PV’s strong innervation by VPm in L4 (based on Sermet et al 2019) that in Fig 5 Sup 1 you don’t see a similar trend in PV cells when comparing Modulation amplitude vs Cell depth. What could the reason for this be?

7. While the sparsity of VIP in deeper layers aids to the difficulty in recording, it would have been nice to see a few more cells in >300um depths.

8. “Periods with intermediate whisker movement activity were classified neither as quiet nor as whisking.” A comparison of Vm dynamics between intermediate and whisking events would be interesting.

9. “Across the four cell classes, we found a large, fast depolarization upon active whisker touch in excitatory and PV neurons, and somewhat delayed, smaller responses in VIP and SST neurons (Figure 6B and C).” This was a bit confusing as in Figure 1C, depolarization and firing seem to precede active touch in VIP cells.

10. In Figure 2 a longer time interval for example traces in A would be helpful, specifically for EXC group.

11. It is unclear and difficult to interpret the type of whisker movement in Figure 3? Is time 0 in B referring to active touch?

12. In Figure 3C and D it is very unclear what the statistic are corresponding to. What is the difference between stats on the x-axis and stats above the graph?

13. In Figure 4 a lot of comparisons between quiet and whisking are made but one thing lacking is the separation by depth. In both the PV group and SST group there is a lot of variability in suppression and enhancement due to whisking. It would be interesting if this was layer-specific. Can the authors please break the data down by layer to explore this?

6. PLOS authors have the option to publish the peer review history of their article (what does this mean?). If published, this will include your full peer review and any attached files.

Reviewer #1: No

Reviewer #2: No

---

## [Author Response · Author response to Decision Letter 0]

21 Apr 2023

Please see separate point-by-point Reviewer Reply file. Below is a copy-paste version, but formatting has been lost.

Reviewer comments cited from the decision email are in Times New Roman italic font.

Our replies are in Arial font. 

Reviewer #1

In this manuscript, Kiritani et al. describe the membrane potential dynamics of multiple classes of excitatory and inhibitory cells across cortical layers in the primary whisker somatosensory barrel cortex (wS1) of awake head-fixed mice during various behavioral conditions (quiet wakefulness, free whisking, and active touch). To this aim, they used whole-cell two-photon guided recordings targeted to fluorescently-labelled neurons in the C2 barrel column while monitoring whisker movements with a camera. They found that the four different classes of recorded neurons had different supra-threshold and sub-threshold dynamics of their membrane potential during quiet wakefulness. Moreover, PV, VIP, and excitatory cells depolarized upon whisking, while SST interneurons hyperpolarized at the onset of whisker movements. The hyperpolarization of the membrane potential of SST cells at the beginning of whisking was a decreasing function of cell depth, being almost absent for deeper SST cells. Upon active whisker touch, excitatory and PV neurons showed pronounced depolarization, while VIP and SST neurons displayed smaller and delayed responses to touch. The author finally analyzed cellular responses to touch as a function of the inter-contact interval (ICI). While excitatory, PV, and VIP cells displayed suppression of the touch-evoked PSP at short ICIs, SST neurons switched response polarity as a function of the ICI (hyperpolarizing at long ICIs and depolarizing at short ICIs).

This study provides a systematic and well-conceived characterization of the supra- and sub-threshold response of different cell types within wS1 in awake head-fixed mice during different behavioral conditions. Results are well presented, the text is clear, and the discussion addresses a sufficient number of critical issues. I support publication of the current version of the manuscript in PLOS One. I only have a couple of very minor suggestions (please find them below).

1) Page 9, line 17 from top, “Many neurons responded with rapid changes…”: please include reference to Figure 6 Figure Supplement 1, so that the reader may quantitatively evaluate how many neurons respond.

We have now included reference to Figure 6 Figure Supplement 1 (now S13 Fig). On lines 510-511, we now write: “Many neurons responded with rapid changes in Vm upon whisker-object contact (Fig 6A and S13A Fig).”

1) Page 16, lines 10-12 from bottom, “.. it is increasingly clear that there are many subclasses of PV, VIP and SST neurons based both on transcriptomic analyses, anatomical features and connectivity”: please include appropriate reference for the transcriptomic, anatomical, and connectivity classification.

On lines 764-767, we now refer to key papers reporting subclasses of GABAergic subgroups: “Equally, it is increasingly clear that there are many subclasses of PV, VIP and SST neurons based both on transcriptomic analyses, anatomical features and connectivity, as well as additional groups of GABAergic neurons not expressing PV, VIP or SST (Muñoz et al., 2017; Prönneke et al., 2015; Tasic et al., 2018, 2016).”  

Reviewer #2

Kiritani et al. use whole-cell patch clamp recordings from both excitatory and inhibitory neurons across the cortical depth to characterize cell-type specific responses to quiet states, free whisking, and active whisker touch. Although prior studies have investigated this using extracellular recordings, the current study expands this work by recording both sub- and suprathreshold responses in both superficial and deep layers using two-photon guided whole-cell recordings to record from excitatory, PV, SST, and VIP neurons. Much of the data had been collected as part of prior studies but reanalyzed here for direct comparisons, and this will be a useful reference in the field. The data provides a good foundation for further investigation into circuit changes driven by other behaviors such as detection or discrimination learning. A major strength of this manuscript is the ability to compare across different brains states which is important for furthering our understanding of sensory processing. However, the authors have not fully taken advantage of their ability to target neurons at multiple depths to compare cell-type specific responses in different layers, except in some supplemental figures which are not well-integrated into the text.

The reviewer is correct to point out that part of the data come from previously published studies (“Much of the data had been collected as part of prior studies”), but in fact the large majority of data presented here are new recordings, never before published. Out of the n = 240 total recordings included in our analysis, n = 192 recordings are new, previously unpublished recordings. We now write this explicitly in the Results section on lines 289-293: “In addition to newly obtained data (n = 192 neurons), here we include new analysis of previously published data from 10 PV neurons and 10 SST neurons (Pala & Petersen, 2018); as well as new analysis of 28 previously published anatomically-identified excitatory neurons (Crochet et al., 2011).”

The reviewer is also correct to point out our limited presentation of depth-dependent properties. This is largely because we only found clear depth-dependence in a small number of our analyses. The depth-dependence is therefore largely presented in Supplementary Figures, where the interested reader can look in more detail. In, addition, we have now added a layer-specific classification of our analyses, to help more simply represent the data, but in many cases there is no obvious pattern that emerges, and therefore the new layer-specific analyses are also presented as Supplementary Figures. 

Major Concerns:

1. The authors justify the need for whole-cell recordings in vivo, but then do not use their data to make any mechanistic inferences. In Figure 3B, the brief hyperpolarization of EXT (and also, it appears, PV) at whisker onset could be aligned to the activity of a specific class of interneuron. What is the most likely candidate? If it is PV neurons, what might this mean? Another exciting possibility enabled by this investigation is that the authors might be able to leverage this labeled dataset to build a classifier for future recording and cell identification. Was this attempted? This opportunity should be acknowledged in the discussion at least.

We agree with the reviewer that we have not attempted to reveal synaptic circuit mechanisms, and it is a clear shortcoming of this study. Most importantly neuronal circuit perturbation experiments are missing. We think it is difficult to speculate given the incompleteness and large heterogeneity of our observations. On the other hand, one strength of this publication is that the raw data will be freely available at Zenodo, so in principle, other investigators could make computational network models to examine possible circuit mechanisms underlying the observed data and try their favorite classification algorithms. On lines 750-753, we now write: “In further analyses of the dataset, it could be interesting to train classifiers based on our cell class-labelled data, which might then be useful to help identify cells recorded under other conditions where no cell class labels are present.”

2. The authors made claims about how “previous extracellular measurements of cell class-specific AP firing in barrel cortex has revealed prominent differences depending upon the depth of the cell body relative to the pial surface (Muñoz et al., 2017; Yu et al., 2019, 2016)” Yet, for the main figures the only data they split and analyzed by depth was SST (Figure 3E). Were there prominent differences in PV responses between layers? Supplemental figures did convey data based on depth but the way it is presented was complicated and sometimes not easy to follow.

The reviewer is correct to highlight that we did not make much progress in understanding depth dependence of Vm dynamics. In fact, high cell-to-cell variability of neuronal properties across depth made it difficult for us to report strong conclusions, beyond the effect, also highlighted by the reviewer, of the SST neurons. This is why the depth-dependent analyses are largely presented in Supplemental figures, which we think is appropriate. In addition to depth dependence, in new Supplemental figures, we now present layer-specific classification of our data to help the reader, but without finding new prominent differences across layers.

3. The analysis is typically focused on very short time periods before and after whisking onset. Can the authors create a histogram with a longer time period of analysis? This may be useful for investigators using stimulation paradigms where responses are not limited to 100 ms, such as learning tasks or delayed match-to-sample tasks.

The reviewer is correct to point out that many of our analyses focus on fast time scales. The whisking cycle is at roughly 10 Hz, so 100 ms is an interesting time scale to examine for phase-locking and active touch responses. Different whisking bouts and active touch sequences have different durations and it becomes less informative to display averaged traces for long time-scales. However, longer periods are considered for free whisking data in Figure 4, where we compare 2-s long periods of quiet vs continuous free-whisking, aiming to investigate changes in cortical states.

4. The trimming of all whiskers but C2, and the use of a single whisker stimulus is highly artificial. Did the authors have recordings where animals were using the entire whisker apparatus? Did this change the response properties of their identified neurons? How long before the recording session was trimming carried out? Were there changes in the response properties of neurons due to trimming?

The reviewer is correct that this is a highly artificial situation for the mice in our study. It would indeed be interesting to investigate differences in behavior and neuronal circuits comparing mice with a full whisker pad. On the other hand, this would add considerable complexity to the whisker tracking and analysis of active touch times of different identified whiskers. Our experiments were therefore restricted to C2 single-whisker mice, and we explicitly point this out in the Methods section on lines 125-126 and lines 169-170. 

Minor Concerns:

1. The authors should indicate what state they analyzed to extract mean firing rates. Were animals locomoting during quiet wakefulness?

We think the reviewer’s comment relates to the data shown in Figure 2. All data in this figure are from periods of quiet wakefulness. Our definition of quiet wakefulness is that the whiskers are not moving. In our experience, if the whiskers are not moving then typically neither are other body parts. Our head-restrained mice are not on a treadmill / other device, so there is no possibility for locomotion. One might consider this as a sort of ‘ground state’ for the awake somatosensory cortex of the mouse. On lines 321-326, we now write: “We first analysed Vm dynamics during quiet wakefulness, which was defined to be periods in which the mouse did not move its whisker (see Materials and methods). Typically, movements of other body parts are also accompanied by whisker movement, so in the quiet state it is likely that there is very little movement of the mouse altogether, defining what one might consider as a ‘ground state’ of neuronal activity in the awake somatosensory cortex.”

2. The authors described in the introduction that a limitation in the current Vm studies is the depth at which they can achieve. Yet, based on what is described in the text it suggests this study is still focused on superficial layers. While the depth correlation analysis found in several supplemental figures was informative, it was not straightforward to read or interpret. Grouping cell properties based on lamination and comparing values would provide a more comprehensive interpretation of some major point being conveyed. For example, in Fig 6 Sup 1 comparing the early vs late responses for L2/3, L4 and L5 cells would provide a more comprehensible view of the data being presented.

The reviewer is correct, that although we present Vm measurements from deeper than most previous recordings, we are nonetheless still biased to the more superficial layers 2-4, with almost no recordings from L5. This is an important limitation of the current study, that we highlight in the text in lines 753-757: “In the future, it will also be of great importance to investigate Vm of neurons in the deeper layers 5 and 6. However, targeting whole-cell recordings through two-photon imaging to such deep neurons is very challenging, and might require the application of new technologies such as three-photon imaging, which promises increased contrast for deep tissue imaging (Ouzounov et al., 2017).” In response to the reviewer’s suggestion, we have now added layer-specific classification of data, as a complement to the depth-dependent analyses. The new layer-specific analyses are presented in a series of new Supplementary figures, but, unfortunately, it remains difficult to find prominent layer-specific differences.

3. What is the difference between Fig 1 E and F? There appears to be a discrepancy in number of cells recorded based on depth. Also why is there no reconstruction of EXC?

In the revised manuscript, we have now tried to clarify what is represented in the two panels. In Fig 1E, we show the depth of the electrophysiological whole-cell recordings. In Fig 1F, we show the distribution of PV-Cre, VIP-Cre and SST-Cre positive cells in the C2 barrel column of example Cre-dependent tdTomato reporter mice from which we did not record. On lines 312-318, we now write: “(E) Distribution of the estimated cell body depths for each recorded neuron according to cell class. Dashed lines indicate depth boundaries used to define cortical layers. (F) Anatomical reconstruction of the cell body locations of GABAergic neurons within the C2 barrel column (top) in three example brains from which we did not record, with the C2 barrel in layer 4 colored green. The distributions of PV-expressing (n=4 mice), VIP-expressing (n=3 mice) and SST-expressing (n=3 mice) neurons are quantified across mice along the depth of the cortical column (below) with the histogram indicating mean ± SD.”

4. The authors report they can achieve depths up to 600um but Figure 1F shows reconstructed cells in 800-1200um depths.

As explained above, Fig 1F shows the distribution of PV-Cre, VIP-Cre and SST-Cre positive cells in the C2 barrel column of a Cre-dependent tdTomato reporter mouse, not the depth of recorded neurons.

5. The statistics and cell numbers were not always easy to read on the graphs. The gray for non-significant samples was also very difficult to read.

We have tried to improve readability by increasing font size and making the gray darker.

6. “Consistent with this, given that VPM provides strong input to L3 and L4 (Sermet et al., 2019), here we found that deeper lying neurons in layers 3 and 4, had larger phase-locked Vm fluctuations compared to more superficial neurons.” It is surprising that given PV’s strong innervation by VPm in L4 (based on Sermet et al 2019) that in Fig 5 Sup 1 you don’t see a similar trend in PV cells when comparing Modulation amplitude vs Cell depth. What could the reason for this be?

The reviewer raises an interesting question. We agree that one might expect PV neurons in L4 to shower higher phase-locked modulation compared to more superficial PV neurons. I do not think we have a good explanation for this. One possibility might be the less specific connectivity of PV neurons compared to EXC neurons, with PV neurons often receiving excitatory synaptic input from a large fraction of nearby neurons. This could tend to average out different phase-locked modulations.

7. While the sparsity of VIP in deeper layers aids to the difficulty in recording, it would have been nice to see a few more cells in >300um depths.

We agree with the reviewer, and, indeed, much more data is needed in order to more fully understand the C2 barrel column.

8. “Periods with intermediate whisker movement activity were classified neither as quiet nor as whisking.” A comparison of Vm dynamics between intermediate and whisking events would be interesting.

Although the reviewer is correct that this is an additional interesting analysis, we think that our figures are already very complex, and we think that introducing more conditions will hinder clarity for the reader. 

9. “Across the four cell classes, we found a large, fast depolarization upon active whisker touch in excitatory and PV neurons, and somewhat delayed, smaller responses in VIP and SST neurons (Figure 6B and C).” This was a bit confusing as in Figure 1C, depolarization and firing seem to precede active touch in VIP cells.

The reviewer points to the complexity of sensory and motor representations in VIP neurons. VIP neurons on average depolarize upon the onset of active whisking. Active touch by definition follows the onset of whisking. In Figure 6B&C, we average across multiple repetitive whisker-object contacts, and therefore the whisking onset depolarization is largely averaged out by the many subsequent active touch events during prolonged bouts of active whisking.

10. In Figure 2 a longer time interval for example traces in A would be helpful, specifically for EXC group.

We already show long time scales in Figure 1. For the interested reader, we will make all data publicly available, so each Vm recording can be examined in detail at short and long time scales.

11. It is unclear and difficult to interpret the type of whisker movement in Figure 3? Is time 0 in B referring to active touch?

Figure 3 does not relate to active touch. This figure examines the onset of free whisking (time = 0 s), in which the mouse voluntarily decides to move its whiskers back-and-forth as if scanning its immediate facial space. No object was placed in the way of the whisker in the free whisking paradigm, therefore no active touches occur during free whisking. We have clarified the figure legend to explicitly point out the absence of the object.

12. In Figure 3C and D it is very unclear what the statistic are corresponding to. What is the difference between stats on the x-axis and stats above the graph?

We thank the reviewer for highlighting this point, which was not described in the previous version of our figure legends. The stats below the x-axis indicate the result of Wilcoxon signed rank tests comparing the data of each cell class to zero. The stats above the graph examine possible differences between cell classes, and were computed using Kruskal-Wallis test followed by a Tukey-Kramer multiple comparison test. On lines 420-427, we now write: “(C) Mean change in Vm at whisking onset (0-200 ms) for the four cell classes. Open circles show individual neuron values. Filled circles with error bars show mean ± SD. Statistical differences for the change in membrane potential comparing quiet and whisking were computed using a Wilcoxon signed rank test for each cell class (shown below the data points). Statistical differences between cell classes were computed using Kruskal-Wallis test (p = 1.1x10 9) followed by a Tukey-Kramer multiple comparison test (shown above the graph). (D) Same as C, but for the change in firing rate (Kruskal-Wallis test, p = 1.9x10 4).”

13. In Figure 4 a lot of comparisons between quiet and whisking are made but one thing lacking is the separation by depth. In both the PV group and SST group there is a lot of variability in suppression and enhancement due to whisking. It would be interesting if this was layer-specific. Can the authors please break the data down by layer to explore this?

We now present layer-specific analyses in new Supplementary figures S10 and S11, but unfortunately no prominent effects emerged.

---

## [Decision Letter · Decision Letter 1]

1 Jun 2023

Membrane potential dynamics of excitatory and inhibitory neurons in mouse barrel cortex during active whisker sensing

PONE-D-22-33584R1

Dear Dr. Petersen,

We’re pleased to inform you that your manuscript has been judged scientifically suitable for publication and will be formally accepted for publication once it meets all outstanding technical requirements.

Kind regards,

Laurens W. J. Bosman, Ph.D.

Academic Editor

PLOS ONE

Additional Editor Comments (optional):

Dear Carl,

Thank you for revising your manuscript. All comments have been properly addressed.

Sorry for the delay during the second round of review, but I am happy that I can suggest to accept this very nice piece of work for publication in PLoS ONE.

Reviewers' comments:

Reviewer's Responses to Questions

**Comments to the Author**

1. If the authors have adequately addressed your comments raised in a previous round of review and you feel that this manuscript is now acceptable for publication, you may indicate that here to bypass the “Comments to the Author” section, enter your conflict of interest statement in the “Confidential to Editor” section, and submit your "Accept" recommendation.

Reviewer #2: All comments have been addressed

2. Is the manuscript technically sound, and do the data support the conclusions?

Reviewer #2: Yes

3. Has the statistical analysis been performed appropriately and rigorously? 

Reviewer #2: Yes

4. Have the authors made all data underlying the findings in their manuscript fully available?

Reviewer #2: Yes

5. Is the manuscript presented in an intelligible fashion and written in standard English?

Reviewer #2: Yes

6. Review Comments to the Author

Reviewer #2: I am satisfied with the revised manuscript. The authors have addressed all reviewer concerns, and the manuscript is now suitable for publication in PLoSOne.

7. PLOS authors have the option to publish the peer review history of their article (what does this mean?). If published, this will include your full peer review and any attached files.

Reviewer #2: No

---

## [Editor Report · Acceptance letter]

6 Jun 2023

PONE-D-22-33584R1 

Membrane potential dynamics of excitatory and inhibitory neurons in mouse barrel cortex during active whisker sensing 

Dear Dr. Petersen:

I'm pleased to inform you that your manuscript has been deemed suitable for publication in PLOS ONE. Congratulations! Your manuscript is now with our production department. 

Kind regards, 

on behalf of

Dr. Laurens W. J. Bosman 

Academic Editor

PLOS ONE